# Elementary effects for models with dimensional inputs of arbitrary type and range: Scaling and trajectory generation

**Rik J. L. Rutjens**[1]*, **Leah R. Band**[1,2]⊕, **Matthew D. Jones**[3]⊕, **Markus R. Owen**[1]⊕

**1** School of Mathematical Sciences, University of Nottingham, Nottingham, United Kingdom, **2** Division of Plant and Crop Sciences, School of Biosciences, University of Nottingham, Loughborough, United Kingdom, **3** School of Geography, University of Nottingham, Nottingham, United Kingdom

⊕ These authors contributed equally to this work.
* pmxrr3@nottingham.ac.uk

## Abstract

The Elementary Effects method is a global sensitivity analysis approach for identifying (un) important parameters in a model. However, it has almost exclusively been used where inputs are dimensionless and take values on [0, 1]. Here, we consider models with dimensional inputs, inputs taking values on arbitrary intervals or discrete inputs. In such cases scaling effects by a function of the input range is essential for correct ranking results. We propose two alternative dimensionless sensitivity indices by normalizing the scaled mean or median of absolute effects. Testing these indices with 9 trajectory generation methods on 4 test functions (including the Penman-Monteith equation for evapotranspiration) reveals that: i) scaled elementary effects are necessary to obtain correct parameter importance rankings; ii) small step-size methods typically produce more accurate rankings; iii) it is beneficial to compute and compare both sensitivity indices; and iv) spread and discrepancy of the simulation points are poor proxies for trajectory generation method performance.

**Data Availability Statement:** All relevant data are within the manuscript and its Supporting information files. In addition, all code and data that support the findings of this study are openly

## 1 Introduction

Models in the biological and environmental sciences typically have many parameters [1–3]. Calibration of these often requires empirical data, which can be costly or simply impossible to obtain (see [1] and the references therein). However, often only a small subset of parameters have a significant influence on a specific system output [4, 5]. As such, it can be beneficial for model development to identify unimportant parameters, so they can be set to a fixed value. Efforts can then be concentrated on accurately estimating the most important factors. This can greatly decrease dimensionality of the model parameter space, while increasing trust in the model. Sensitivity analysis (SA), the study of how uncertainty in the model output can be attributed to the different sources of uncertainty in the model inputs, is a common tool for this [2, 6].

But what is sensitivity exactly? In a local context an unambiguous definition is readily available for continuous deterministic models in terms of partial derivatives: given an output $y$ dependent on inputs $x_1, \ldots, x_n$, the local sensitivity of $y$ to an input $x_i$ at a point $\mathbf{x}^\star$ in the

available at https://github.com/pmxrr3/EE_theory_2023.

**Funding:** The authors received no specific funding for this work.

**Competing interests:** The authors have declared that no competing interests exist.

parameter space is given by

$$s_i = \frac{\partial y}{\partial x_i}\bigg|_{x*},$$ (1)

supposing $y$ is differentiable at $\mathbf{x}^\star$. Here, $s_i$ is also known as the sensitivity coefficient, and characterizes the independent effect of $x_i$, when all other inputs are held constant. If the inputs and outputs are dimensional, the sensitivity coefficient tends to be scaled in one of two ways [1, 7, 8]. Multiplying by a ratio of reference values $x_i^0/y^0$ yields a relative sensitivity index. This index is normalized and enables comparisons between factors with different units or values at different orders of magnitude, but it fails to account for the variability in the input and output [1]. Alternatively, multiplying by the ratio of standard deviations $\sigma_x/\sigma_y$ gives a variance sensitivity index, but this approach requires information about the spread of each input and output. Local interaction effects are typically defined in a similar way to first-order local sensitivity, by considering mixed higher order partial derivatives. In general, interaction effects can be defined at different scales, and no single method or scale is capable of fully characterizing interactions in numerical simulators [9].

One-at-a-time (OAT) methods, changing one parameter at a time from a fixed base point and assessing the effect on the model output, are commonly used local sensitivity analysis techniques. This assessment may be by use of (discretized) derivatives (i.e. Eq (1)), or might simply involve visual inspection of the model outputs [6]. Although OAT methods are still popular, it has been suggested that local SA methods may only prove informative in very specific situations (e.g. inverse problems, or approximating a model output in a small region of output space) [10]. In general, OAT is therefore not recommended for rigorous SA; global sensitivity analysis methods (GSA) should be used instead [10, 11].

A variety of approaches have been proposed for GSA, leading to different notions of (global) sensitivity [2, 12]. Sensitivity is often described as *the influence of a parameter on a model output* (see e.g. [13]), but the precise form of 'influence' is not always stated. The 'correct' notion may vary on a case-to-case basis, depending on the specific goal one wants to achieve. For example, consider the simple output $Y = X_1 + X_2$, where $X_1$ takes values in [0, 10] and $X_2$ in [100, 101] uniformly. $X_2$ contributes most to the mean magnitude of the output $Y (= 5 + 100.5)$, so it could be argued this parameter is most important. Alternatively, $X_1$, having a larger range, has the most significant contribution to the variability in the output (variance of $Y = \frac{1}{12}(10^2 + 1^2)$). In this work the following **notion of global sensitivity** is considered, which is the prevalent one in GSA: *the sensitivity of output $Y_j$ to input parameter $X_i$ is the relative contribution of the variability in the input parameter to the variance of the output*. The (finite) range of an input is used here as input variability. Other notions of input and output variability can be used, e.g. mean, standard deviation or interquartile range.

Although SA typically considers continuous inputs, inputs may also be discrete. For example, one may have integer-valued inputs representing different scenarios. Sensitivity for models with categorical (i.e. discrete) input variables has been considered for some approaches, such as variance-based SA [14, 15]. For a detailed treatise of GSA, including methods that take the complete distribution of the output into account, we refer to [1–4, 11, 16, 17] and the references contained therein.

Here we focus on the Elementary Effects method (EE) [18], a qualitative screening method for (un)important parameters, where the space of model outputs is characterized by a relatively low number of strategically placed simulation points. From these points, finite differences (called *Elementary Effects*) can be calculated as a measure of how the output changes when one input changes. Finally, by aggregating these effects for each combination of input and output,

measures of (global) sensitivity of the outputs for the inputs are obtained. EE is also capable of detecting non-linearity or interaction effects.

As far as we are aware, descriptions of the Elementary Effects method (see e.g. [18–26]) assume models are dimensionless with inputs taking real values on the unit interval. However, it is commonplace in practice (and in many environmental models) for models to have dimensional outputs, and for inputs to take values on arbitrary intervals or of different types (real, integer or Boolean). We therefore discuss the necessary changes to make EE applicable to general models (Sec. 2, 3), whilst reviewing popular and recent improvements to EE. In particular, necessary and sufficient scalings of the elementary effects to prevent erroneous rankings are discussed, and two versions of a scaled dimensionless sensitivity measure are introduced (Sec. 4). In addition, we investigate through numerical experiments what trajectory generation method and sensitivity measure are generally best for EE, and consider to what extent spread and discrepancy of the set of simulation points can be used as proxies for trajectory generation method performance (Sec. 5).

## 2 Elementary effects method

### 2.1 Original formulation (extended to general models)

Let $X_i$, $i = 1, \ldots, k$ be dimensional input parameters with units $[X_i]$, taking values in $[\min_i, \max_i]$ uniformly. If the parameter can only take integer values, it takes values in the set $\{\min_i, \min_i + 1, \ldots, \max_i\}$. The same holds for Boolean parameters, but then $\min_i = 0$ and $\max_i = 1$, where 0 encodes false and 1 stands for true. $x_i$ denotes the dimensionless equivalent scaled to the unit interval, i.e.

$$x_i = \frac{X_i - \min_i}{\max_i - \min_i}, \qquad (2)$$

henceforth referred to as *scaled dimensionless* parameters. The assumption of uniformly distributed inputs can be relaxed to include arbitrary distributions. Scaling the sampled parameter values from $[0, 1]$ to the $[\min_i, \max_i]$-interval should then be done using the corresponding inverse cumulative density function (CDF). The dimensional outputs of interest are denoted by $Y_j$, $j = 1, \ldots, q$ with corresponding unit $[Y_j]$.

The parameter points used for the analysis are sampled from a regular discrete subset of the complete scaled dimensionless parameter space (typically called $\Omega \subset [0, 1]^k$) containing $p_i$ regularly spaced points in the $x_i$-direction, and are then transformed to the actual parameter space. $p_i$ is also called the number of levels for parameter $x_i$. The scaled dimensionless parameter $x_i$ thus takes values in the set

$$x_i \in \left\{ \frac{j}{p_i - 1} : j = 0, 1, \ldots, p_i - 1 \right\}, \qquad (3)$$

see Fig 1, while (using Eq (2)) the actual parameter value is an element of

$$X_i \in \left\{ \min_i + \frac{j(\max_i - \min_i)}{p_i - 1} : j = 0, 1, \ldots, p_i - 1 \right\}. \qquad (4)$$

This formulation restricts the choice of parameter bounds and number of levels for Boolean and integer parameters; for the former one must set $p_i = 2$, while for the latter, the following

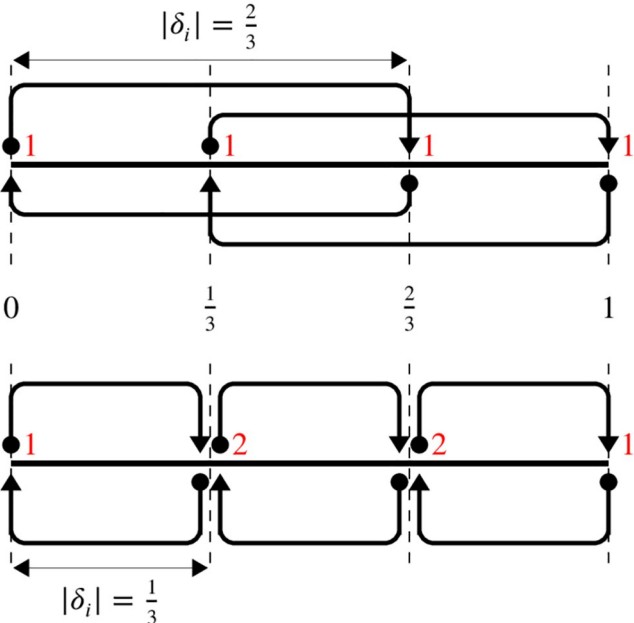

**Fig 1. Schematic representation of parameter sampling probabilities for a parameter $x_i$ on [0, 1] with $p_i$ = 4 levels.**
The starting point of an arrow (circle) represents a sampled parameter value, the end point (triangle) represents the perturbed value. Together they lead to an Elementary Effect (e.g. Eq (7)). Equal sampling probability means that each discrete parameter value has an equal number of incoming arrows (indicated in red). Above: optimal choice for the step size $|\delta_i| = p_i/[2(p_i - 1)] = 2/3$ leads to equal sampling probabilities. Below: non-optimal choice for $|\delta_i|$ leads to a higher probability of sampling interior points.

relation must be satisfied to ensure integer parameter values:

$$\max_i - \min_i = m(p_i - 1), \quad \text{for some } m \in \mathbb{N}. \tag{5}$$

If a parameter $x_i$ takes values on the unit interval, an *elementary effect* of $x_i$ on an output $Y_j$ is given by the finite difference

$$ee_{ij}^n = \frac{Y_j(x_1, \ldots, x_{i-1}, x_i + \delta_i, x_{i+1}, \ldots, x_k) - Y_j(\mathbf{x})}{\delta_i}, \tag{6}$$

where $\mathbf{x} = (x_1, \ldots, x_k)$. Here, the superscript $n$ is an index to distinguish different $\mathbf{x}$, to emphasize that the elementary effect can be calculated at numerous points in the parameter space. $\delta_i$ is a predetermined value in the set $\{\pm 1/(p_i - 1), \pm 2/(p_i - 1), \ldots, \pm 1\}$ such that $x_i + \delta_i$ still lies in [0, 1] (see S1 Appendix in S1 File). Morris [18] argues the optimal value for the *step size* $|\delta_i|$ is $p_i/[2(p_i - 1)]$, where $p_i$ is chosen to be even. This ensures equal sampling probabilities for all discrete parameter values, as shown in Fig 1. Note that this may necessitate the use of parameter-dependent values of $\delta$ and $p$; for Boolean parameters, one must choose $p_i = 2$, $\delta_i = 1$, but for real and integer inputs a higher number of levels (hence $\delta_i \neq 1$) is typically preferred. In some cases, e.g. for an integer input parameter $x_i$ with $\min_i = 1$ and $\max_i = 3$, Eq (5) shows it is not possible to use the optimal value for $\delta$ as $p_i$ must be odd. To our knowledge, the current literature assumes a fixed (even) value of $p$ and the optimal value for $\delta$ for all parameters.

For dimensional inputs and arbitrary input ranges we introduce the following generalized form of the elementary effect:

$$EE_{ij}^n = \frac{Y_j(X_1, \ldots, X_{i-1}, X_i + \Delta_i, X_{i+1}, \ldots, X_k) - Y_j(\mathbf{x})}{\Delta_i}. \qquad (7)$$

Here $\Delta_i = (\max_i - \min_i)\delta_i$. The effect $EE_{ij}^n$ given by Eq (7) is dimensional with units $[EE_{ij}^n] = [Y_j]/[X_i]$.

Note that the dimension of $\Delta_i$ is equal to the dimension of $X_i$. Hence, even if all $p_i$'s and all input parameter ranges are equal, thereby equalizing the magnitude of each $\Delta_i$, one should still refrain from dropping the index, because the *units* of the $\Delta_i$'s might be different.

The total number of elementary effects associated with input $X_i$ (and output $Y_j$) is equal to the number of parameter points for which $x_i \leq 1 - |\delta_i|$. Those are the points for which an increase by $|\delta_i|$, the other point needed for the calculation of an effect, still lies in the parameter space (see Fig 1). There are $p_i - |\delta_i|(p_i - 1)$ discrete values that $x_i$ may take that fulfil $x_i \leq 1 - |\delta_i|$ (all values except those larger than $|\delta_i|$) and

$$\prod_{\substack{j = 1, \ldots, k \\ j \neq i}} p_j \qquad (8)$$

combinations for the other parameter values, so the total number of elementary effects for input $X_i$ is equal to

$$(p_i - |\delta_i|(p_i - 1)) \prod_{\substack{j = 1, \ldots, k \\ j \neq i}} p_j. \qquad (9)$$

This result reduces to the one in [18] ($p^{k-1}[p - |\delta|(p - 1)]$) if all the $p_i$'s and $\delta_i$'s are equal. The goal in the original formulation is to estimate the distributions of these effects for each combination of input and output. Following Morris [18], these distributions are denoted by $F_{ij}$, where the first index depicts the input and the second the output. If there is only one output under consideration, we simply write $F_i$. It is generally not feasible (nor desirable) to calculate every possible effect; for $k = 50$ input parameters and $p = 4$ and $\delta = 2/3$ for each input, this would amount to $\sim 10^{29}$ simulations (Eq (9)). Instead, the goal is to generate a small set of $Q = r(k + 1)$ simulation points, typically $Q \sim 1000$, that still provide good coverage of the parameter space. Each $F_{ij}$ is then characterized by its sample mean and sample standard deviation over $r$ effects [18] (see Sec. 3 for more detail).

## 2.2 Trajectory generation using an optimized winding stairs design and fixed step sizes

The most naive way of sampling a set of $r$ effects for each of the $k$ inputs would be to randomly sample $r$ base points in $\Omega$. Since the calculation of each effect requires two output values, this would require a total number of $2rk$ simulations. However, by generating trajectories in parameter space (Figs 2 and 3) and using each point (except for the start and end) for the calculation of not one, but two effects, the number of required simulations decreases to $r(k + 1)$. This approach is an example of a *'winding stairs'* design (Fig 2). Alternatively, one can use a *'radial design'* (Fig 4), leading to star-shaped trajectories in parameter space (described further in Section 2.3).

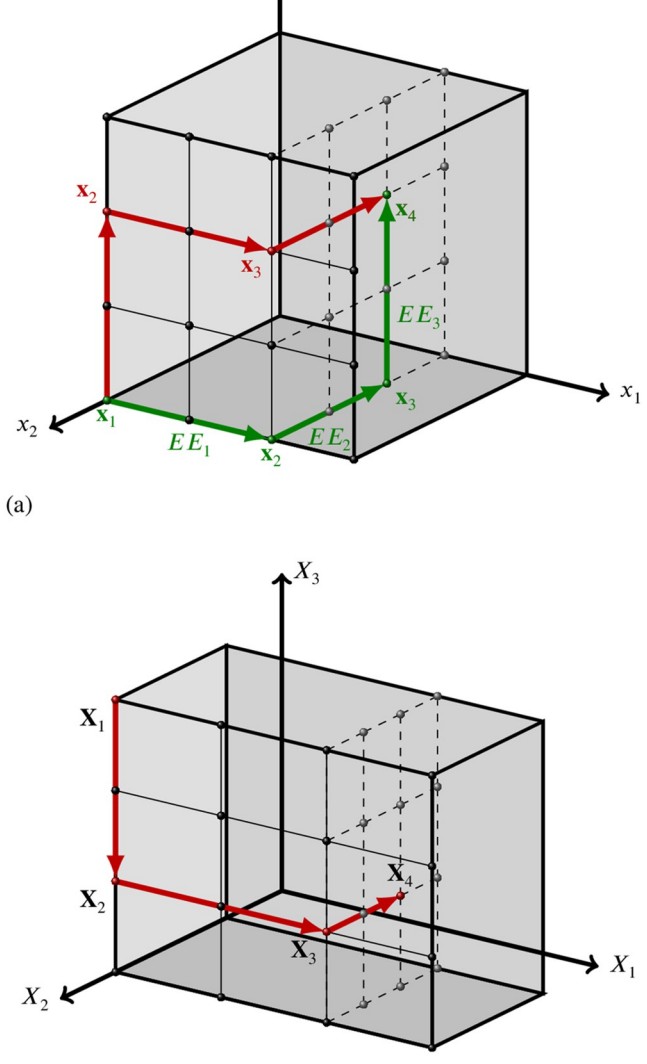

**Fig 2. Realisation of trajectories in a winding stairs design with (red) and without (green) random column permutations.** Here, $k = 3$ and we set $p = 4$ and $\delta = 2/3$ for all parameters. Effects are calculated using both endpoints of the arrows, as indicated in green. a) Trajectories in the discrete unit hyperspace $\Omega$. Green is without column permutation, red is with column permutation. b) Trajectory (with column permutation) in the actual parameter space. $X_1$, $X_2$ and $X_3$ take values in [0, 4/3], [1/3, 1] and [0, 1], respectively.

The following is an adaptation of the description by Morris [18]; notation differs slightly, and we account for allowing parameter-dependent step sizes $\delta_i$. All calculations in this section are done on $\Omega$, the discrete unit hypercube, using the scaled dimensionless quantities ($x_i$, $\delta_i$, etc.). After a trajectory is generated on $\Omega$, one simply transforms it to the actual parameter space using Eq (2). A winding stairs trajectory is a semi-random walk through $\Omega$ which has the following properties:

- there is exactly one value change in each dimension;

- the value in dimension $i$ changes from $x_i$ to $x_i + \delta_i$ or vice versa, with equal probability;

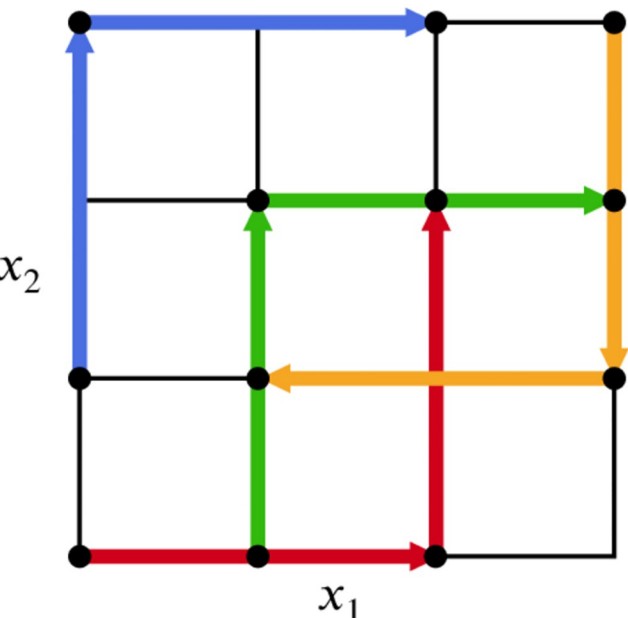

**Fig 3. Example of an optimal set of trajectories for $k = 2, r = 4$, $p = 4$ and $|\delta| = 2/3$ for both parameters.**

- the order in which dimension steps are taken is semi-random. Starting with the order sequence $[x_1, x_2, \ldots, x_k]$ (meaning the first step is in the $x_1$-direction, the second in the $x_2$-direction, and so forth), all elements are randomly permuted, but only with other elements that have the same corresponding $\delta_i$ and $p_i$. For example, if we have 3 input parameters, where $\delta_1 = \delta_2 \neq \delta_3$ and $p_1 = p_2 \neq p_3$, there are two (2!) possible sequence orders: $[x_1, x_2, x_3]$ and $[x_2, x_1, x_3]$.

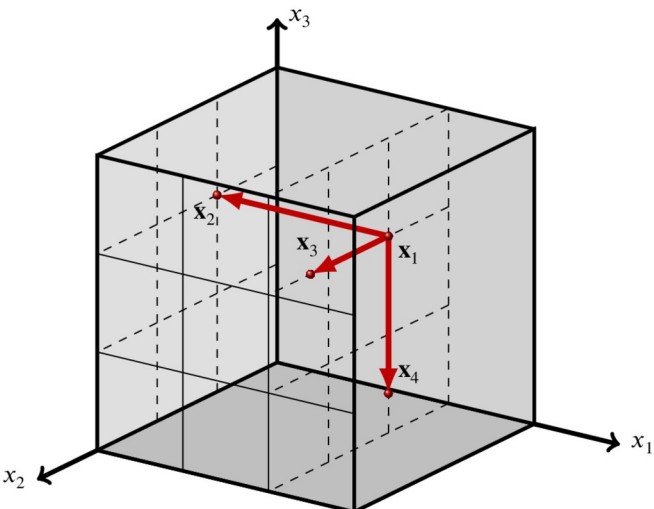

**Fig 4. Radial design sample in the unit cube with $k = 3$ parameters.**

A winding stairs trajectory $T_n$ can be fully characterized by the matrix $\mathbf{B}_n^\star$, given by:

$$\mathbf{B}_n^\star = \left( \mathbf{J}_{k+1,1} \mathbf{x}_{\text{init}} + \frac{1}{2} \left[ \mathbf{B}_0 \mathbf{D} + \mathbf{J}_{k+1,k} \right] \text{diag}(|\boldsymbol{\delta}|) \right) \mathbf{P}. \tag{10}$$

Row $j$ represents the $j$-th scaled dimensionless parameter point in the trajectory, while column $i$ refers to the value of scaled dimensionless parameter $x_i$. $\mathbf{J}_{n,m}$ is the $n \times m$ matrix of ones, $\text{diag}(|\boldsymbol{\delta}|)$ is the $k \times k$ diagonal matrix containing the (scaled dimensionless) step sizes $|\delta_i|$ and $\mathbf{P}$ is a $k \times k$ random column permutation matrix in which columns $i$ and $j$ may be permuted only if the corresponding parameters have the same number of levels, i.e. if $p_i = p_j$. $\mathbf{D}$ is a $k \times k$ diagonal matrix where each diagonal element is either +1 or -1 with equal probability. $\mathbf{x}_{\text{init}}$ is a $1 \times k$ row vector containing an initial scaled dimensionless parameter point, randomly sampled from the restricted subset of $\Omega$ denoted by $\{0, 1/(p_i - 1), \ldots, 1 - |\delta_i|\}^k$ (to ensure that $x_i + |\delta_i|$ still lies in the unit interval), where the power denotes a Cartesian product. Finally, the $(k + 1) \times k$ matrix $\mathbf{B}_0$ is given by

$$\mathbf{B}_0 = \begin{bmatrix} -1 & -1 & -1 & \cdots & -1 \\ 1 & -1 & -1 & \cdots & -1 \\ 1 & 1 & -1 & \cdots & -1 \\ 1 & 1 & 1 & \ddots & -1 \\ \vdots & \vdots & \vdots & \ddots & -1 \\ 1 & 1 & 1 & \cdots & 1 \end{bmatrix}. \tag{11}$$

Note that $\frac{1}{2}[\mathbf{B}_0 \mathbf{D} + \mathbf{J}_{k+1,k}]\text{diag}(|\boldsymbol{\delta}|)$ is a $(k + 1) \times k$ matrix where column $i$ is given by either

$$\left. \begin{bmatrix} |\delta_i| \\ \vdots \\ |\delta_i| \\ 0 \\ \vdots \\ 0 \end{bmatrix} \right\} i \text{ times} \quad \text{or} \quad \left. \begin{bmatrix} 0 \\ \vdots \\ 0 \\ |\delta_i| \\ \vdots \\ |\delta_i| \end{bmatrix} \right\} i \text{ times} \tag{12}$$

with equal probability and independent of the other columns, i.e. if we ignore the random permutation matrix $\mathbf{P}$, the value of scaled dimensionless parameter $i$ would either start at $\mathbf{x}_{\text{init},i}$ or $\mathbf{x}_{\text{init},i} + |\delta_i|$, change to $\mathbf{x}_{\text{init},i} + |\delta_i|$ or $\mathbf{x}_{\text{init},i}$ after the $i$-th step in the trajectory, respectively, and then remain the same in the rest of the steps. Thus, the permutation matrix randomly changes the sequence of parameter steps. Fig 2 depicts two trajectories, without and with random column permutations. Multiple trajectories can be put in matrix form by simply concatenating the $r$ $\mathbf{B}_n^\star$-matrices:

$$\tilde{\mathbf{B}} = \begin{bmatrix} \mathbf{B}_1^\star \\ \mathbf{B}_2^\star \\ \vdots \\ \mathbf{B}_r^\star \end{bmatrix}. \tag{13}$$

The question now is how to create a good coverage of the parameter space with a relatively low number of trajectories. Campolongo et al. [19] introduced a simple but effective strategy

called *'optimized trajectories'*, (OT) which is described here, as it is commonly used: $M$ random trajectories are generated, typically $M \approx 1000$, and the subset of size $r \ll M$ with the largest dispersion in the input space is selected. Typically, $r$ ranges between 4–20 (likely due to historical computational limitations), although recent papers [27, 28] indicate it might be worthwhile to increase this number at the expense of additional runtime. The notion of *spread* of a set of trajectories is defined via the following measure of distance between two trajectories:

$$d_{ml} = \begin{cases} \sum_{i=1}^{k+1} \sum_{j=1}^{k+1} \|\mathbf{x}_i^m - \mathbf{x}_j^l\|_2, & \text{for } m \neq l; \\ 0 & \text{otherwise,} \end{cases} \tag{14}$$

where $\mathbf{x}_i^m$ denotes the $i$-th point of the $m$-th trajectory, i.e. $d_{ml}$ is the sum of the geometric distances between all the couples of points of two trajectories [19]. The optimal set of trajectories is then found using a brute force approach, by considering the measure of spread given by

$$D_{k_1 \ldots k_r} = \sum_{\substack{i,j=0 \\ i \neq j}}^{r} d_{k_i k_j}^2 \tag{15}$$

for all combinations of $r$ trajectories out of $M$, denoted by the vector of trajectory indices ($k_1$, ..., $k_r$), where $k_i \in \{1, \ldots, M\}$ and $k_1 < \ldots < k_r$. Campolongo et al. [19] use the square root of this quantity, but that does not affect the location of the maximum. Finally the set of $r$ trajectories with the highest value of $D_{k_1 \ldots k_r}$ is selected. We simply call this maximal value $D$ in what follows, i.e.,

$$D = \max[D_{k_1 \ldots k_r}; \{k_1, \ldots, k_r\} \subset \{1, \ldots, M\}]. \tag{16}$$

Fig 3 depicts a realisation of the optimized trajectories approach containing $r = 4$ trajectories for $k = 2$ scaled dimensionless parameters with $p = 4$ levels. This process is computationally taxing, since it involves calculating $\binom{r}{2}$ distance measures $d_{ml}$ and the spread $D_{k_1 \ldots k_r}$ $\binom{M}{r}$ times. Especially the latter figure quickly becomes prohibitively large; for $M = 1000$ and $r = 25$, $\binom{1000}{25} \sim 10^{49}$ values of $D_{k_1 \ldots k_r}$ need to be calculated. Little computer memory is needed, since only the maximum value $D$ thus far and the new value of $D_{k_1 \ldots k_r}$ need to be stored, and optimal sets of trajectories can be generated beforehand and independently of the actual model simulations. Nevertheless, the brute force approach is not feasible in practice. Khare et al. [21], although they use the tag 'OT', actually employ a different method, which we call Efficient Optimized Trajectories (EOT): for each of $M$ initially generated trajectories, generate a set of $r$ trajectories by successively adding those with the highest spread w.r.t. to those already in the set. This leads to $M$ sets of $r$ trajectories from which the set with the highest total spread is selected. In algorithmic terms:

```
EOT:
Step 1: Generate M trajectories {T¹, ..., Tᴹ}
Step 2: for i = 1 to M
          Set S₁ⁱ = Tⁱ;
          for j = 2 to r
            Sⱼⁱ = Tᵏ, where k = argmaxₙ₌₁,...,ₘ(D(S₁ⁱ,...,Sⱼ₋₁ⁱ, Tⁿ));
          end
          Save spread Dᵢ = D(S₁ⁱ,...,Sᵣⁱ).
        end
Step 3: Pick the set of trajectories {S₁ⁱ,...,Sᵣⁱ} with the highest spread
Dᵢ.
```

**Table 1. Runtime for generating the set of trajectories for different combinations of *k* (number of parameters) and *r* (number of trajectories).** †: $M = 200$ trajectories. *: initial pool of $M = 500$ trajectories. 'n/a': these take many weeks to complete (extrapolated from the $k = 10$, $r = 4$ case) and are thus not shown. 'Standard' Sobol Radial is representative for all QR-based approaches as listed in Section 5. All computations were done on a HP Zbook Studio G4 computer.

| | | Runtime (seconds) | | | |
|---|---|---|---|---|---|
| *k* | *r* | OT† | EOT† | EOT* | 'Std' Sob. rad. |
| 10 | 4 | 14 | 2.9 | 11 | $<0.001$ |
| 20 | 8 | n/a | 5.5 | 25 | 0.001 |
| 50 | 20 | n/a | 21 | 165 | 0.002 |
| 100 | 40 | n/a | 91 | 1130 | 0.006 |

This produces a local spread maximum, which may be less than the global maximum, but greatly reduces computational cost; see Table 1. With this approach, we were able to replicate the computation times reported in Fig 3 of [21]. While none of the well-known papers (such as [19, 29]) explicitly mention this more efficient algorithm, it is likely that most papers have in fact employed EOT instead of the brute force OT approach (Ruano et al. [20] being the exception). We employ EOT in this study.

In (E)OT, the trajectory starting points are sampled randomly. Selecting these points through Latin Hypercube Sampling (LHS) or a quasi-random (QR) sequence should increase the spread and coverage of the *r* selected trajectories. However, exploratory numerical experiments (not shown here) showed no significant change compared to a random sample (see also [19]). Apparently, the benefits of generating a large pool of *M* trajectories outweigh those of LHS or QR sampling.

## 2.3 Trajectory generation using a radial design and a QR-sequence (extended to general models)

A popular alternative to generating trajectories in a 'winding stairs' approach is to use a radial design [30]. The key difference is that in a radial design steps are taken from the same base point (Table 2). This approach is essentially *r* OAT-designs with different base points. Each point unequal to the base point differs in exactly one (unique) coordinate from the base point (Fig 4). An important difference between the radial design as described here and in e.g. [29, 30] and the standard winding approach for EE ((E)OT) is that the former no longer makes use of fixed step sizes $|\delta_i|$. Instead, a step size may take any value in $(0, x_i]$ (step to the left) or $(0, 1 - x_i]$ (step to the right), and step sizes in the same direction may differ in magnitude for

**Table 2. Radial sampling design versus winding stairs sampling design without random column permutations.** *k* inputs are considered here, resulting in $k + 1$ points in parameter space. The base point is given by $(a_1, a_2, \ldots, a_k)$. In OT the $a_i$ are elements from a discrete set and $b_i = a_i \pm |\delta_i|$, whereas in the radial design (as in [30]) $a_i$ and $b_i$ can take any value in $[0, 1]$. Table adapted from [30].

| Radial | Point | Winding |
|---|---|---|
| $a_1, a_2, a_3, \ldots, a_k$ | $\mathbf{x}_1$ | $a_1, a_2, a_3, \ldots, a_k$ |
| $b_1, a_2, a_3, \ldots, a_k$ | $\mathbf{x}_2$ | $b_1, a_2, a_3, \ldots, a_k$ |
| $a_1, b_2, a_3, \ldots, a_k$ | $\mathbf{x}_3$ | $b_1, b_2, a_3, \ldots, a_k$ |
| $a_1, a_2, b_3, \ldots, a_k$ | $\mathbf{x}_4$ | $b_1, b_2, b_3, \ldots, a_k$ |
| ... | ... | ... |
| $a_1, a_2, a_3, \ldots, b_k$ | $\mathbf{x}_{k+1}$ | $b_1, b_2, b_3, \ldots, b_k$ |

different base points. As such, the number of levels $p_i$ are obsolete in this method. The steps $\delta_i$ (now a function of the specific trajectory) are not predefined, but calculated a posteriori.

To ensure a uniform distribution of the $r$ base points in the parameter space, a quasi-random (QR) or low-discrepancy sequence is typically used; see e.g. [29, 30] for examples in radial EE, and [31, 32] and Section 5.1 for more on QR sequences in general. QR sequences are designed to produce point sets that cover a space both efficiently (i.e. with a low amount of points) and evenly (i.e. approximating a uniform distribution).

Sobol sequences are the most popular choice of QR sequence. These sequences use polynomials over the field $\mathbb{Z}_2$ to form successively finer uniform partitions of the unit interval and then reorder the coordinates in each dimension. To initialize the algorithm, a set of so-called direction numbers is required; we use those provided by Joe and Kuo [33]. The built-in function SobolSequenceGenerator in the Apache Commons Math 3.6.1 Java library is used to generate QR vectors. We do not give a detailed description here, but refer to [33–35], Chapter 5, for details.

We also consider the recently presented $R_d$ sequences [36], which may have more favourable properties of rapid and uniform coverage [36, 37]. As far as we are aware, this sequence has not been used in GSA so far. The $R_d$ sequence in $k$ dimensions $\{\mathbf{z}_n\}_{n\in\mathbb{N}_+}$ is simply given by

$$\mathbf{z}_n = \alpha_0 + n\boldsymbol{\alpha} \bmod 1, \quad n = 1, 2, 3, \ldots, \tag{17}$$

where $\alpha_0$ is a fixed offset ($\frac{1}{2}$ in this work) and

$$\boldsymbol{\alpha} = \left(\frac{1}{\phi_k}, \frac{1}{\phi_k^2}, \ldots, \frac{1}{\phi_k^k}\right),$$

in which $\phi_k$ is the unique positive root of the generalized golden ratio equation

$$x^{k+1} = x + 1.$$

For numerical purposes, $\phi_k$ can either be estimated through Newton iteration, or by truncating the identity

$$\phi_k = \sqrt[k+1]{1 + \sqrt[k+1]{1 + \sqrt[k+1]{1 + \sqrt[k+1]{1 + \ldots}}}}.$$

The QR sequence of choice is used to generate a set of $r$ base points (as $r \times k$ matrix $A$) and a set of $r$ perturbation vectors ($r \times k$ matrix B). In principle, there are two ways of filling the matrices $A$ and $B$. One can generate a $r \times 2k$-matrix, where each row is an element of a $2k$-dimensional QR sequence, and subsequently set $A$ to be the left half of the matrix and $B$ the right half. This is the correct approach, and is used in [29, 30] and in this work. Alternatively, one can generate a $2r \times k$-matrix by concatenating $2r$ elements of a $k$-dimensional QR sequence, and use the top half for $A$ and the bottom half for $B$. However, we found that this leads to erratic and non-converging behavior (errors in preliminary tests (not shown here) did not decrease with increasing $r$), so this approach should be avoided. It is not exactly clear why this happens, but it is likely to be related to the fact that subsequent points in a QR sequence (hence, subsequent rows in $A$ and $B$) are dependent. In the case of Sobol QR, Campolongo et al. [29] note it might be worthwhile to use a shifted perturbation vector, i.e. to generate a

$(r + q) \times 2k$-matrix, and match base point $A_i$ ($i = 1, \ldots, r$) with perturbation vector $B_{i+q}$, i.e.,

$$\begin{pmatrix} \mathbf{A}_1 & & \mathbf{B}_1 \\ \vdots & & \vdots \\ \mathbf{A}_{1+q} & & \mathbf{B}_{1+q} \\ \vdots & & \vdots \\ \mathbf{A}_{r+q} & & \mathbf{B}_{r+q} \end{pmatrix}.$$

In particular $q = 4$ is reported to give "good results" [29], which is adopted in this work. Nevertheless, in Sobol QR it may happen that one of the elements of the perturbation vector coincides with its base point counterpart, i.e. $A_{ij} = B_{i+q,j}$ for some $j = 1, \ldots, k$. The row vector $[A_{i+q}, B_{i+q}]$ is discarded and regenerated when this happens.

For integer and Boolean inputs, arbitrary step sizes are not allowed, since they may lead to non-integer/Boolean sampling points (Eq (5)). To ensure allowable base points, we propose using the number of levels $p_i$ and step size $|\delta_i|$ from the OT approach to pin the base point coordinates for integer/Boolean inputs to a discrete value as in Eq (3) and then use corresponding step size $|\delta_i|$. That is, given a base coordinate $x_i$ in [0, 1] generated by a QR sequence, we transform the coordinate to a discrete value $\tilde{x}_i$ by:

$$\tilde{x}_i = \begin{cases} 1 & \text{if } x_i = 1; \\ \frac{\lfloor p_i x_i \rfloor}{p_i - 1} & \text{else.} \end{cases} \tag{18}$$

While this quantity is not necessarily integer/Boolean, the coordinate in the actual parameter space is (using Eq (5)):

$$\begin{aligned} X_i &= (\max_i - \min_i)\tilde{x}_i + \min_i \\ &= n(p_i - 1)\tilde{x}_i + \min_i \\ &= \begin{cases} \max_i & \text{if } x_i = 1; \\ n\lfloor p_i x_i \rfloor + \min_i & \text{else,} \end{cases} \end{aligned}$$

for some $n \in \mathbb{N}$, which is an integer/Boolean. Pinning the perturbed coordinate does not work, because there is a high probability ($1/p_i$) the pinned perturbed coordinate will coincide with the base point coordinate (leading to a step size of 0 and an undefined elementary effect in that direction). For example, if $p = 4$ the base coordinate $x_i = 0.3$ will be pinned to $\tilde{x}_i = 1/3$, but any perturbed coordinate in [1/4, 1/2] will be pinned to this same value. Therefore, we step with fixed step size $|\delta_i|$ (or $|\Delta_i|$) (in the direction that keeps the perturbed point in the parameter space).

To distinguish the sampling strategies described here when testing them in Section 5, we refer to the radial design where all points are generated with a Sobol QR sequence by *'standard' Sobol radial* and the equivalent using an $R_d$ QR sequence by *'standard' $R_d$ radial*. Corresponding winding designs are indicated by the postfix *winding* instead of *radial*: *'standard' Sobol winding* and *'standard' $R_d$ winding*. Moreover, as a computationally efficient alternative to EOT (see Table 1), one could use QR sequences to generate the base points, and then transform them regardless of type (real, integer, Boolean) to a discrete value as in (3) (for a given chosen $p_i$) and step with fixed step size $|\delta_i|$ in a radial or winding design, as described above. These approaches are denoted by the prefix *'pinned'* instead of *'standard'*.

## 2.4 Alternative approaches to trajectory generation

To further address the computational expense of the combinatorial optimization problem in Campolongo's optimal trajectory strategy [19], to enhance uniform coverage of the parameter space and to further increase accuracy in sensitivity rankings, a number of alternative sampling designs have been proposed, discussed here for completeness.

Ruano et al.'s modified optimal trajectories scheme (MOT) [20] checks only a subset of all possible trajectory sets, leading to a considerable reduction of computation time (compared to OT and EOT) but at the cost of deteriorating parameter space coverage (compared to both OT and EOT) [21]. Khare et al. [21] introduce Sampling for Uniformity (SU), which aims to generate simulation points that are close to the asserted input parameter distributions, whilst also maximizing trajectory spread. SU outperforms EOT and MOT in some benchmark tests on computation time, uniformity and screening effectiveness, but scores lower on maximizing trajectory spread. Khare et al. [21] also list a number of older approaches. For a number of recent approaches (including cluster sampling), the reader is referred to [26, 38–41] and the references therein.

## 3 Sensitivity measures

In the original formulation, each distribution of effects $F_{ij}$ (as presented in Sec. 2) is characterized by its sample mean and sample standard deviation over $r$ effects [18]:

$$\mu_{ij} = \frac{1}{r} \sum_{n=1}^{r} EE_{ij}^n; \tag{19}$$

$$\sigma_{ij} = \sqrt{\frac{1}{r-1} \sum_{n=1}^{r} \left(EE_{ij}^n - \mu_{ij}\right)^2}. \tag{20}$$

A large magnitude of the mean $\mu_{ij}$ indicates a great influence of input $X_i$ on output $Y_j$, while a large standard deviation $\sigma_{ij}$ indicates substantial interaction terms and/or non-linearity are present in output $Y_j$. While this may not provide a full characterisation of the distribution (e.g. if it is not symmetric), the typical low number of observations ($r \sim 6 - 20$) [19–21] generally prohibits more detailed specification. Campolongo et al. [19] proposed to also consider the mean of the absolute effects, $\mu_{ij}^\star$ to filter out potential cancelling of terms:

$$\mu_{ij}^\star = \frac{1}{r} \sum_{n=1}^{r} |EE_{ij}^n|. \tag{21}$$

This measure has become one of the most prevalent, and is used in one of our new measures (Sec. 4.4).

In recent years, a number of alternative sensitivity measures or ways to aggregate effects have been proposed. These aim to provide more stable results (i.e. fewer changes in parameter importance ranking as the number of trajectories ($r$) is varied), allow for different interpretations of the effects, or produce results that better align with the notion of sensitivity. Menberg et al. [28] obtained more stable ranking results by using the median value of the absolute effects, $\chi_{ij}$, instead of the mean (Eq (21)). The idea is that this measure is less sensitive to outliers (or a lack thereof) if the effects have a skewed and/or long-tailed distribution, since the number of effects per input parameter in EE is typically low [19–21]. Saltelli et al. [11] argue that one should always take the scaled dimensionless step size $\delta_i$ ($\in [0, 1]$) instead of the actual step size $\Delta_i$ ($\in [\min_i, \max_i]$) to calculate elementary effects (reiterated in 2018 by Feng et al.

[42]). This amounts to a multiplication of the effect in Eq (7) by $\max_i - \min_i$. A more detailed treatise of effect scaling is given in Section 4. To remove output scale effects, Wang et al. [43] introduce the normalized absolute effect for dimensionless parameters

$$P_{ij}^n = \frac{|EE_{ij}^n|}{\sum_{l=1}^k |EE_{lj}^n|},$$

(22)

where the normalization is over the inputs at the $n$-th trajectory. This leads to a normalized global sensitivity index for the $i$-th parameter given by

$$\tau_{ij} = \frac{1}{r}\sum_{n=1}^r P_{ij}^n = \frac{1}{r}\sum_{n=1}^r \frac{|EE_{ij}^n|}{\sum_{l=1}^k |EE_{lj}^n|}.$$

(23)

In other words, $\tau_{ij}$ is obtained by first normalizing effects and then averaging over trajectories. Finally, by averaging over different outputs, Wang et al. [43] argue a measure for the average sensitivity of a parameter on multiple output variables is found as:

$$\beta_i = \frac{1}{J}\sum_{j=1}^J \tau_{ij} = \frac{1}{rJ}\sum_{j=1}^J\sum_{n=1}^r \frac{|EE_{ij}^n|}{\sum_{l=1}^k |EE_{lj}^n|},$$

(24)

where $J$ is the number of outputs. We would argue the use of this last measure is questionable for most practical applications; suppose two parameters exhibit opposite sensitivities for two outputs, being very sensitive for one output but not sensitive for the other, the average measure would attribute a moderate importance to both factors, while in practice both are important. Alternatively to Eq (23), Wu [40] first averages the absolute effects, and then normalizes these quantities, leading to the relative importance evaluation index for dimensionless parameters

$$S_{ij} = \frac{\mu_{ij}^\star}{\sum_{l=1}^k \mu_{lj}^\star} = \frac{\sum_{n=1}^r |EE_{ij}^n|}{\sum_{l=1}^k \sum_{n=1}^r |EE_{lj}^n|},$$

(25)

where $\mu_{ij}^\star$ is the mean of the absolute effects (Eq (21)). To determine what parameters are important and unimportant (for a given output $Y_j$), the $S_{ij}$'s are sorted in ascending order leading to a sequence $S_{i_1 j} < S_{i_2 j} < \ldots < S_{i_k j}$. The $q$ non-influential parameters (*inactive variables* in [40]) are then those for which

$$\sum_{m=1}^q S_{i_m j} < \frac{h}{100}, \quad \sum_{m=1}^{q+1} S_{i_m j} \geq \frac{h}{100},$$

(26)

where $h$ is a predefined percentage, e.g. 30%. A higher (lower) unimportance threshold $h$ therefore leads to more (less) unimportant parameters. Influential parameters on a given output (*active variables* in [40]) are those with a relative importance evaluation index above a predetermined threshold $S_{0j}$. Following [40],

$$S_{0j}(h) = \widehat{\mu}_{0j} + 3\widehat{\sigma}_{0j}$$

(27)

is used, where $\widehat{\mu}_{0j}$ and $\widehat{\sigma}_{0j}$ are the sample mean and standard deviation of the $q$ $S_{ij}$'s

corresponding to inactive variables, (i.e. variables that correspond to $S_{i_mj}$ that satisfy Eq (26)). Thus,

$$\widehat{\mu}_{0j} = \frac{1}{q}\sum_{m=1}^{q} S_{i_mj}; \tag{28}$$

$$\widehat{\sigma}_{0j} = \sqrt{\frac{1}{q-1}\sum_{m=1}^{q}\left(S_{i_mj} - \widehat{\mu}_{0j}\right)^2}. \tag{29}$$

This means that the importance threshold $S_{0j}$ is a function of the $q$ unimportant $S_{ij}$'s, and consequently a function of the threshold $h$ through Eq (26). This ensures a significant difference between important and unimportant parameters. Changing the importance threshold $S_{0j}(h)$ (given $h$), e.g. by increasing (decreasing) the number of standard deviations, will decrease (increase) the number of important parameters, but does not influence the number of unimportant parameters. Both Wu [40] and Wang et al. [43] do not consider the standard deviation of the effects $\sigma_{ij}$, but purely base parameter importance rankings on mean effects. Finally, several papers use the ratio of (absolute) mean and standard deviation of the effects instead of their separate magnitudes to characterize parameter sensitivity, e.g. [20, 28, 44]. As an example, Menberg et al. [28] define four regions: if for a given output $\mu_{ij}^\star/\sigma_{ij} < 0.1$, the effects are linear, if $0.1 < \mu_{ij}^\star/\sigma_{ij} < 0.5$ they are considered monotonic, $0.5 < \mu_{ij}^\star/\sigma_{ij} < 1$ constitutes the 'almost monotonic' region and finally if $\mu_{ij}^\star/\sigma_{ij} > 1$, the effects are non-linear and/or non-monotonic. Yang et al. [44] consider parameters to have non-linear effects if $|\mu_{ij}/\sigma_{ij}| > 2/\sqrt{r}$, where $r$ is the number of independent samples for each parameter. We do not use these ratios in this work.

## 4 Scaling of effects

The Elementary Effects method as described above works well for dimensionless models where all inputs take values in [0, 1]. In practice, however, many models are dimensional and/or their inputs take values on non-unit intervals. This may lead to erroneous ranking results, as is shown in the examples below. To alleviate this issue, the effects must be scaled or the model must be made dimensionless. The latter is not always a feasible option, especially in biology or environmental sciences where models may have over 100 parameters. In that case, it is difficult to find all the dimensionless quantities, and even if those are known, it may be difficult to translate the sensitivity of the dimensionless quantities back to sensitivity of the original parameters. In this section, we present new results to demonstrate what types of scaling work and which do not.

Following Sin and Gernaey [27], we split the scaling of the effects in two, considering separately scaling in the input- and output direction. We denote the scaled effects by

$$\widehat{EE}_{ij}^{n} = EE_{ij}^{n}\frac{c_{x_i}}{c_{y_j}}, \tag{30}$$

where $c_{x_i}$ and $c_{y_j}$ are the scaling of model factors and outputs, respectively. Since the mean (median) of the scaled absolute effects is the same as the scaled mean (median) of the absolute effects (e.g. for the mean $\frac{1}{r}\sum_{n=1}^{r}|\widehat{EE}_{ij}^{n}| = \mu_{ij}^\star \cdot c_{x_i}/c_{y_j}$), we will simply write the latter in what follows.

## 4.1 Scaling effects in $X_i$-direction

The necessity of scaling effects in the $X_i$-direction in dimensional models, or in models where parameters take values on non-unit intervals, becomes evident with the following two examples. Firstly, let $Y(X_1, X_2) = X_1^2 + X_2$ be the output of interest. Assume both $X_1$ and $X_2$ are dimensional. It follows that their dimensions are $[X_1]$ and $[X_1]^2$, respectively. By definition of an elementary effect we have

$$
\begin{aligned}
EE_1^n \quad &= \frac{Y(X_1 + \Delta_1, X_2) - Y(X_1, X_2)}{\Delta_1} \\
&= \frac{(X_1 + \Delta_1)^2 + X_2 - (X_1^2 + X_2)}{\Delta_1} \\
&= \Delta_1 + 2X_1
\end{aligned}
\tag{31}
$$

and similarly

$$
EE_2^n = \frac{Y(X_1, X_2 + \Delta_2) - Y(X_1, X_2)}{\Delta_2} = 1,
\tag{32}
$$

where $X_1$ and $X_2$ are arbitrary values. The first effect has dimension $[X_1]$, so the magnitude of the effect depends on the chosen units. On the other hand, the effect of the second parameter is a dimensionless constant. The same holds for the measures by which parameters are typically ranked (effect mean (Eq (19)) and standard deviation (Eq (20))): $\mu_2 = 1$ and $\sigma_2 = 0$, while the measures for $X_1$ depend on the units of that parameter. In other words, if one does not scale the effects, one can choose units such that $\mu_1 \ll \mu_2$ (e.g. km), making it appear like a parameter is relatively unimportant, but one can just as well select units such that the opposite is true (e.g. mm), indicating the parameter is in fact the most important one.

Secondly, let $Y = X_1 + X_2$, with $X_1 \in [0, 20]$ and $X_2 \in [9, 11]$, so that the inputs have equal mean but different standard deviation. Clearly $X_1$ contributes most significantly to the variability in the output. However, the unscaled effects for both parameters equal 1. Only by scaling can we obtain results consistent with our notion of sensitivity; see Table 3.

## 4.2 What scaling to use in $X_i$-direction?

The following does not require dimensionality of the model, but only supposes that the inputs take values on non-unit intervals.

Again let $Y = X_1 + X_2$, with $X_1 \in [0, 20]$ and $X_2 \in [9, 11]$ uniformly. Take $p_1 = p_2 = 4$ and $|\delta_1| = |\delta_2| = 2/3$ and consider the 4 trajectories as depicted in Fig 3. As is shown in Table 3, scaling by the distributional mean of the input and the distributional standard deviation of the input or parameter range results in different rankings. Scaling by the distributional mean gives that both parameters contribute equally to the output mean, when our notion of sensitivity

**Table 3. The right scaling must be chosen so that the results agree with the notion of sensitivity.** EE applied to $Y = X_1 + X_2$, with $X_1 \in [0, 20]$ and $X_2 \in [9, 11]$, $p_1 = p_2 = 4$ and $|\delta_1| = |\delta_2| = 2/3$. Four trajectories are considered, as depicted in Fig 3. The effects are scaled by $c_{x_i}^{(1)} = (\max_i + \min_i)/2$, $c_{x_i}^{(2)} = ([(\max_i - \min_i + 1)^2 - 1]/12)^{1/2}$ or $c_{x_i}^{(3)} = \max_i - \min_i$. In all cases $\sigma_i = 0$.

| | | | $EE_i^n \cdot c_{x_i}^k, k = 1, 2, 3$ | | | | | |
| | $EE_i^n$ | | $c_{x_i}^{(1)}$ (input mean) | | $c_{x_i}^{(2)}$ (input std) | | $c_{x_i}^{(3)}$ (input range) | |
| | $i = 1$ | $i = 2$ | $i = 1$ | $i = 2$ | $i = 1$ | $i = 2$ | $i = 1$ | $i = 2$ |
| (Scaled) elementary effect (Eq (7)), $\mu_i$ (Eq (19)) or $\mu_i^\star$ (Eq (21)) | 1 | 1 | 10 | 10 | 6.06 | 0.82 | 20 | 2 |

should require that $Y$ is more sensitive to $X_1$. On the other hand, scaling by the distributional standard deviation or input parameter range does show that $X_1$ contributes most significantly to the variation in the output. We thus conclude, based on the examples presented here and existing literature (e.g. [11, 42]), that scaling by a function of the input range $(c_{x_i} = c_{x_i}(\max_i - \min_i))$ gives the desired results. The simplest such scaling is $c_{x_i} = \max_i - \min_i$, which is used here. There might be cases where another scaling might be preferred. For instance, if the input parameter is normally distributed ($\sim \mathcal{N}(\mu, \sigma)$), it might make more sense to scale by the standard deviation. Alternatively one could think about how to systematically set $\max_i/\min_i$, e.g. as $\mu \pm 2\sigma$. This is not further explored in this work. Importantly, there is a significant drawback of scaling by a property of the input distribution, thereby making the effects directly dependent on this property, when there is uncertainty about that distribution.

## 4.3 Scaling effects in output direction

Scaling in the output direction does not affect relative results for a given output, since it just amounts to a multiplication of all effects by the same constant. Reasons for scaling nevertheless are i) to non-dimensionalize the effects and/or ii) to normalize the effects or measures to enable comparisons between outputs. The key difference between scaling in the input and output direction is that the inputs have a known (albeit assumed) distribution, while the outputs have an unknown distribution. This means that one can use sample-independent scalings for the inputs, such as the range, distributional mean or standard deviation, which is desirable when constructing a consistent sensitivity measure. For the outputs, one is limited to scalings that depend on sample-dependent values. One can think of mean or standard deviation of the model outputs at the sampled points, or the difference between smallest and largest output value across the sampled space. Alternatively, scalings based on empirical data may exist; for a crop model, biomass could be scaled by the mean biomass from field trials, but also this type of scaling is sample-dependent. As such, sensitivity measures involving scaling in the output-direction are best avoided. In the setting of Eq (30) this is equivalent to taking

$$c_{y_j} = 1, \tag{33}$$

which is done in this work.

## 4.4 A new sensitivity measure

Many of the alternative measures described in Section 3 (e.g. those proposed in [40, 43]), involve averaging or normalisation over elementary effects or effect measures. While this may be logical for dimensionless models, the summation of quantities with potentially different dimensions such as those in Eqs (22)–(25), cannot be interpreted. Moreover, most measures mentioned in Section 3 lack any scaling of the effects, potentially leading to erroneous ranking results. We therefore propose a synthesis of existing measures, resulting in a scaled, dimensionless and normalized measure agreeing with our notion of sensitivity, whilst preventing erroneous ranking results in dimensional models or models with inputs of arbitrary type and range. Taking either the mean of the absolute effects ($\mu_{ij}^\star$; Eq (21)) or the median of the absolute effects ($\chi_{ij}$, following [28]), we scale (following [11]) by

$$c_{x_i} = \max_i - \min_i, \tag{34}$$

and normalize (following [40]), leading to the sensitivity measures

$$S_{\mu^\star}(i,j) = \frac{\mu_{ij}^\star c_{x_i}}{\sum_{l=1}^{k} \mu_{lj}^\star c_{x_l}}, \tag{35}$$

or

$$S_{\chi}(i,j) = \frac{\chi_{ij} c_{x_i}}{\sum_{l=1}^{k} \chi_{lj} c_{x_l}} \tag{36}$$

respectively. Note that $[\chi_{ij} c_{x_i}] = [\mu_{ij}^\star c_{x_i}] = [Y_j]$. Hence, the measures (35) and (36) are dimensionless, independent of scaling in the output direction and are consistent with our notion of sensitivity. Furthermore, the measures take values in [0, 1] and sum to unity (for each output). This allows for the standardized way of identifying the (un)important parameters as described by Wu [40] (Eq (26)). Note that the measures (35) and (36) resemble a discretized version of the differential importance measure introduced by Borgonovo and Apostolakis [45]. Fig 5 shows a visualization of this approach for an example set of 50 sensitivity indices under different unimportance levels (i.e. $h$-values in Eq (26)).

We do not use the standard deviation of the effects in this work, but instead focus on the median or mean of the absolute effects. Nevertheless, an interesting open question is how one should integrate the standard deviation into the analysis. One could for example consider a quantity analogous to the normalized sensitivity measures in Eqs (35) and (36):

$$S_{\sigma}(i,j) = \frac{\sigma_{i,j} c_{x_i}}{\sum_{l=1}^{k} \sigma_{l,j} c_{x_l}}. \tag{37}$$

where $\sigma_{i,j}$ is as in Eq (20). The question is how one should reconcile the two rankings ($S_{\mu^*}(i,j)$ or $S_{\chi}(i,j)$ and $S_{\sigma}(i,j)$), especially when parameters score high on one but low on the other. The work by Borgonovo and Rabitti [46] might be of interest, as they show $\sigma_{ij}^2$ is a biased estimator of the Sobol total sensitivity index (in the case of fixed step sizes). We leave this question for further research.

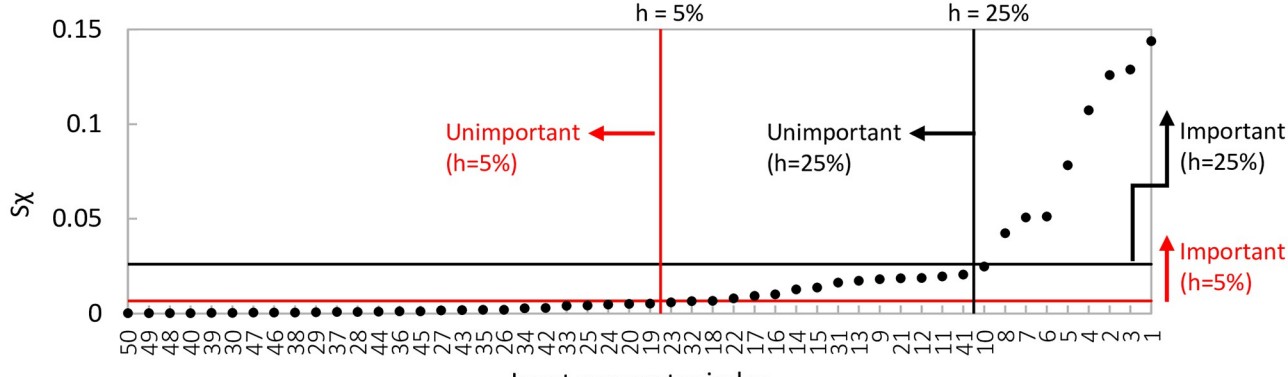

**Fig 5. Visualisation of (un)important parameters and ranking.** Considered here is an example set of sensitivity indices for 50 input parameters. Parameters are ordered based on $S_{\chi}$ (Eq (36)). Vertical lines show the unimportance threshold for $h$ = 5 and 25% using Eq (26), i.e. all parameters to the left of said line are unimportant. Horizontal lines show the corresponding importance thresholds $S_0(h) = \widehat{\mu}_0 + 3\widehat{\sigma}_0$ (Eqs (28) and (29)) for these $h$-values, i.e. parameters above this line are deemed important. Note that for $h$ = 5%, all parameters are either important or unimportant; for $h$ = 25%, there is one parameter that is neither.

## 5 Comparing trajectory generation strategies & sensitivity measures

In this section we investigate which trajectory generation method, in combination with which of the sensitivity measures (Eqs (35) and (36)), is best for EE. Nine trajectory generation methods can be distilled from Section 2. First, five winding stairs designs:

w1. EOT (Enhanced optimized trajectories as employed by Khare et al. [21]);

w2. 'standard' Sobol winding (for real-valued inputs: generate all points by Sobol QR [33]; for integer/Boolean type inputs: generate base value by Sobol QR, transform to nearest discrete value as in Eq (3), then step by $\delta_i$ in a winding design);

w3. 'pinned' Sobol winding (for all inputs: generate base value by Sobol QR, transform to nearest discrete value as in Eq (3), then step by $\delta_i$ in a radial design);

w4. 'standard' $R_d$ winding (w2., but with $R_d$ QR sequence);

w5. 'pinned' $R_d$ winding (w3., but with $R_d$ QR sequence);

and secondly four radial designs:

r1. 'standard' Sobol radial (w2., but using a radial design);

r2. 'pinned' Sobol radial (w3., but using a radial design);

r3. 'standard' $R_d$ radial (r1., but with $R_d$ QR sequence [36]);

r4. 'pinned' $R_d$ radial (r2., but with $R_d$ QR sequence).

Saltelli et al. [30] showed that 'standard' Sobol radial is the better strategy (compared to 'standard' Sobol winding) when estimating the Sobol total sensitivity index (S3 Appendix in S1 File, Eq. (S19)) for a selected set of test functions with $k = 10$ input parameters. Campolongo et al. [29] showed for a maximum of 20 factors (and $r = 2$–8) that 'standard' Sobol radial is also more accurate than OT in identifying (un)important parameters. In Section 5.3 and 5.4 we extend these results by estimating Sobol total senstivity indices and computing parameter rankings, respectively.

Furthermore, even though most trajectory generation approaches are based on maximizing spread and/or minimizing discrepancy (i.e. maximizing uniform parameter space coverage), the relation between spread/discrepancy and ability to correctly rank parameters, identify (un) important factors or calculate sensitivity indices has not yet been ascertained as far as we are aware. We therefore investigate to what extent spread and discrepancy can be used as proxies for sampling technique performance. We do this by calculating the spread and discrepancy of the set of simulation points generated by several trajectory generation methods, and comparing these with the results in Section 5.3 and 5.4.

We do not need to compare all 9 trajectory generation strategies in all experiments. First of all, w3 and w5 ('pinned' Sobol/$R_d$ winding) are extremely similar to w1 (EOT), the only difference being the way the base points are sampled, hence we only look at w1 (whenever computationally feasible) in what follows and assume the results hold for w3 and w5 as well. Secondly, to compare the performance of designs using Sobol sequences versus those using $R_d$ sequences, it suffices to include only a subset of variants; here w2, r1, r2 and r3 are considered. To summarize, in Section 5.1–5.4 below we consider trajectory generation methods w1 (whenever computationally feasible), w2, r1, r2 and r3. In addition we include r4 in Section 5.1 and w4 in Section 5.3.

### 5.1 Spread and discrepancy of sampling strategies

The spread $D$ (Eqs (14 and 15)) measures how far apart **trajectories** are, but does not necessarily indicate how well the **points** uniformly cover the parameter space. Discrepancy is a quantity that measures the uniformity of finite point sets [31]. It originated in the field of QR sequences (or low-discrepancy sequences), where the goal is to generate sequences with high uniformity. More information is provided in S2 Appendix in S1 File. In this work we use the $L_2$-based wrap-around discrepancy $W_2$ [47, 48], given in closed form by

$$
\begin{aligned}
W_2^2(N, k) \quad &= -\left(\frac{4}{3}\right)^k \\
&+ \frac{1}{N^2} \sum_{n=1}^{N} \sum_{m=1}^{N} \prod_{i=1}^{k} \left(\frac{3}{2} - \left[\left|x_i^{(n)} - x_i^{(m)}\right| \cdot \left(1 - \left|x_i^{(n)} - x_i^{(m)}\right|\right)\right]\right),
\end{aligned}
\tag{38}
$$

where $N$ denotes the number of points and $k$ is the dimension of the parameter space. The lower the discrepancy, the better the set of points covers the space uniformly.

For sampling strategies w1, w2, r1, r2, r3 and r4, the spread (14) and (15) and discrepancy (38) are calculated for various combinations of $k$ (the number of input parameters) and $r$ (the number of trajectories). In particular, we vary $r$ between 4 and 100 while fixing $k$ to 50 (Fig 6), and vary $k$ between 10 and 150 while fixing $r$ to 6, 10, 20 or 35 (Fig 7). This covers the range of numbers of inputs and trajectories which are used in practice whilst being computationally feasible. Runtime restricts the number of initial trajectories in EOT (especially for large $k$); in Fig 6 the pool therefore contains $M = 500$ elements, while in Fig 7 it contains only $M = 200$ elements; preliminary experiments (not shown here) showed a negligible difference in spread and discrepancy upon enlarging this pool.

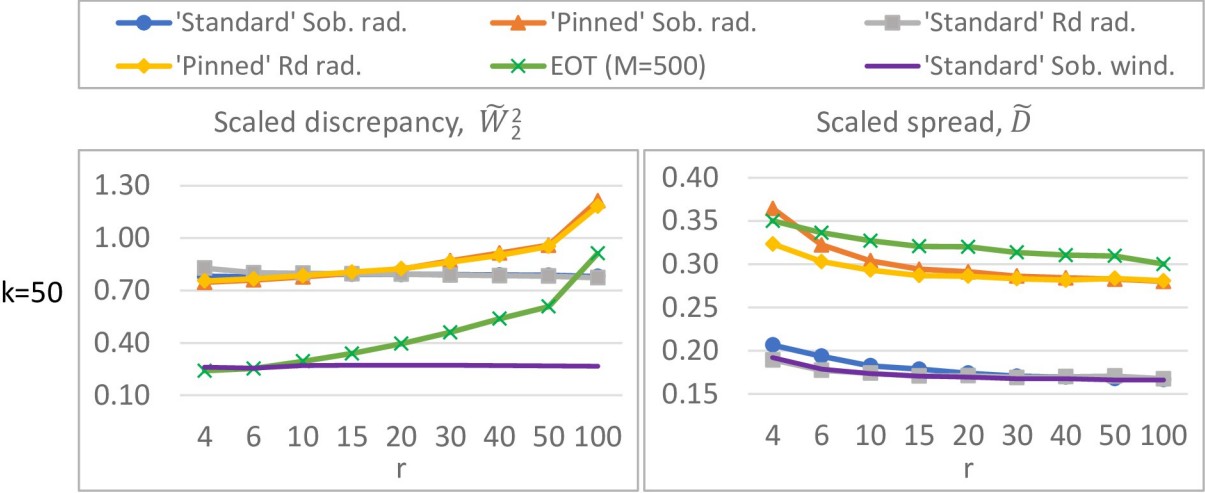

**Fig 6. Discrepancy $\tilde{W}_2^2$ (left) and spread $\tilde{D}$ (right) for different sampling strategies.** The number of model inputs $k = 50$, while the number of trajectories $r$ varies. 'Standard' Sobol winding has the overall lowest discrepancy, EOT has the overall highest spread.

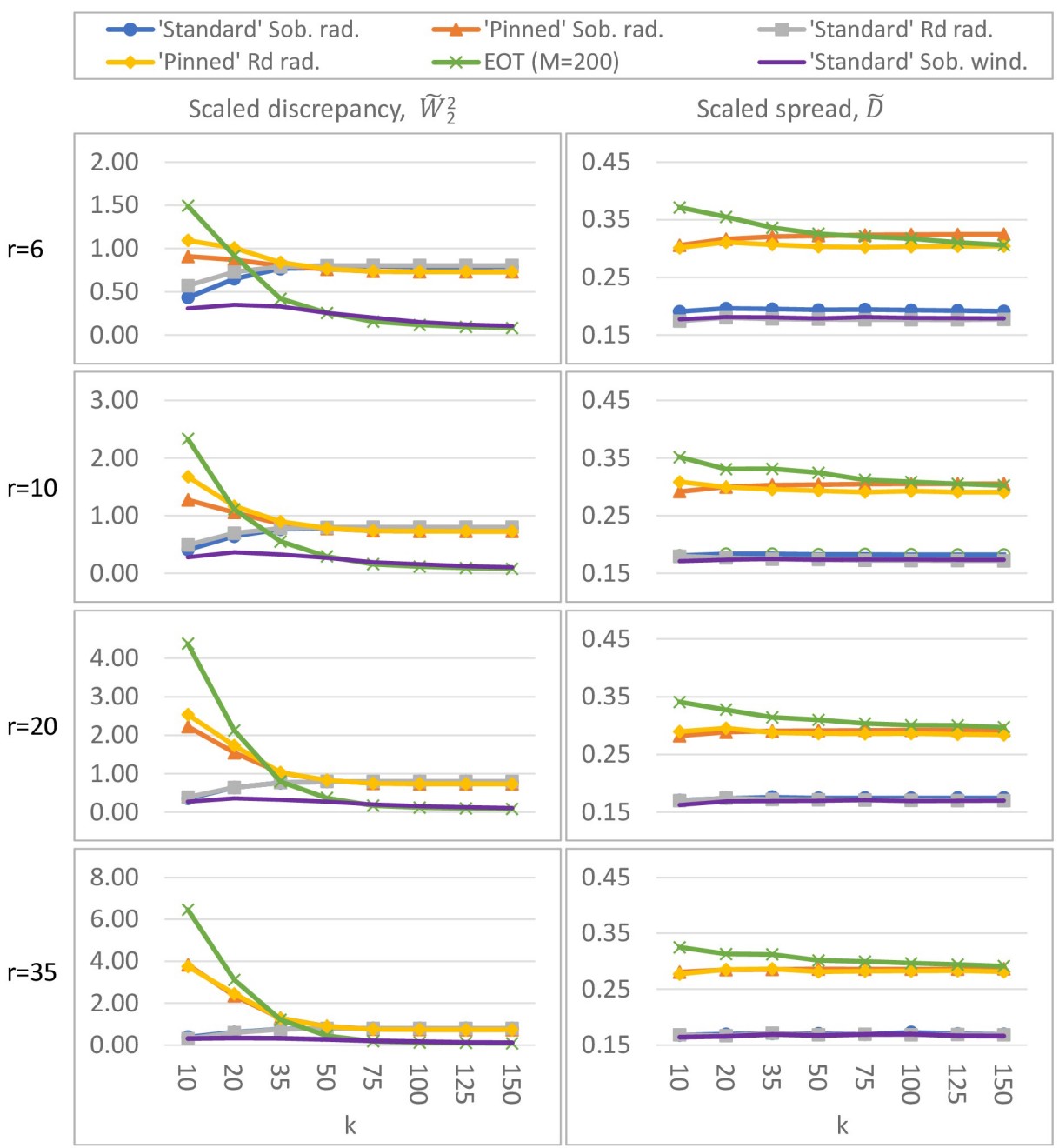

**Fig 7. Discrepancy $\tilde{W}_2^2$ (left) and spread $\tilde{D}$ (right) for different sampling strategies.** In these figures, the number of model inputs $k$ varies, while the number of trajectories $r$ is fixed. 'Standard' Sobol winding has the overall lowest discrepancy, EOT and 'pinned' Sobol/$R_d$ radial have the overall highest spread.

The spread (Eq (16)) is scaled by the number of elements in its expression, i.e.,

$$\tilde{D} = \frac{D}{\binom{r}{2}k[(k+1)^4 + (k+1)^2]}. \tag{39}$$

The discrepancy (Eq (38)) is scaled by the number of points in a trajectory times the expected squared discrepancy of a random uniform sample of size $r(k + 1)$, i.e.,

$$\tilde{W}_2^2(r(k+1),k) \quad = \frac{rW_2^2(r(k+1),k)}{\left(\frac{3}{2}\right)^k - \left(\frac{4}{3}\right)^k}$$

$$= \frac{W_2^2(r(k+1),k)}{(k+1)\mathbb{E}[W_2^2(\mathcal{U}_{r(k+1)},k)]}. \tag{40}$$

These scalings ensure that all results are the same order of magnitude.

**Results.** Our simulations (Figs 6 and 7) reveal the following orderings for spread and discrepancy:

**Ordering of sampling strategies based on spread:** The different sampling strategies are ordered (from largest to smallest spread) as follows: 1) EOT; 2) 'pinned' Sobol/$R_d$ radial; and with a significant margin 3) 'standard' Sobol/$R_d$ radial and 'standard' Sobol winding.

**Ordering of sampling strategies based on discrepancy:** 'standard' Sobol winding is always among the strategies with smallest discrepancy. The ordering of the other sampling strategies depend on the number of input factors $k$ (and to a lesser extent on the number of trajectories $r$). Nevertheless, for sufficiently large $k$ ($\gtrsim 50$), they are ordered (from smallest to largest discrepancy) as follows: 1) 'standard' Sobol winding & EOT; 2) all others. For low $k$ ($\lesssim 15$), they are ordered as follows: 1) 'standard' Sobol winding; 2) 'standard' Sobol/$R_d$ radial 3) 'pinned' Sobol/$R_d$ radial; 4) EOT. In the intermediate range for $k$, 'standard' Sobol winding is the strategy with the lowest discrepancy, with the other techniques following in a $k$- and $r$-dependent order.

The scaled discrepancy (for fixed $r$ and varying $k$) seems to exhibit limiting behavior as $k$ grows large (Fig 7).

## 5.2 Test functions

In the subsequent two sections, four test functions are used. The $K$- and $G^*$-functions are two commonly used dimensionless test functions, and are considered in Saltelli et al. [30], whose experiment we revisit. The six-dimensional test function $f_6$ has been previously presented in [3, 16]. The Penman-Monteith equation [49] is a dimensional equation describing evapotranspiration.

The $K$-function with $k$ inputs is given by

$$K(\mathbf{x}) = \sum_{i=1}^{k}(-1)^i\prod_{j=1}^{i}x_j, \tag{41}$$

where $x \in [0, 1]^k$ uniformly. The $G^*$-function is given by

$$G^*(\mathbf{x}, \mathbf{a}, \boldsymbol{\alpha}, \boldsymbol{\eta}) = \prod_{i=1}^{k}\frac{(1+\alpha_i)|2(x_i+\eta_i - I[x_i+\eta_i]) - 1|^{\alpha_i} + a_i}{1+a_i}, \tag{42}$$

where $I[\cdot]$ is the integer part, $a_i, \alpha_i > 0$ and $\eta_i \in [0, 1]$ for $i = 1, \ldots, k$. The $x_i$ are assumed to be uniformly distributed in [0, 1]. Table 4 lists the values for $\mathbf{a}$ and $\boldsymbol{\alpha}$ for different $k$. The $K$-function contains less non-linearity than the $G^*$-function. Likewise, the low-dimensional versions (i.e. with $k = 10$) are more 'difficult' than their high-dimensional counterparts (i.e. $k = 75$ for the $K$-function and $k = 50$ for the $G^*$-function) because the additional parameters are all relatively unimportant; for the $G^*$-function, this is in part due to the choice of constants in Table 4. As both test functions only contain multiplications of inputs, this has a dampening

**Table 4. Values of a and $\alpha$ for the $G^*$-function (Eq (42)) for different numbers of inputs $k$.**

| $k$ | a | $\alpha$ |
|---|---|---|
| 10 | {0, 0.1, 0.2, 0.3, 0.4, 0.8, 1, 2, 3, 4} | $\alpha_i = 2$ for $i = 1, \ldots, 10.$ |
| 50 | {0, 0.1, 0.2, 0.3, 0.4, 0.8, 1, 2, 3, 4, 0, 0.1, 0.2, 0.3, 0.4, 0.8, 1, 2, 3, 4, 0, 1, 2, 3, 4, 8, 10, 20, 30, 40, 0, 2, 4, 6, 8, 16, 20, 40, 60, 80, 0, 5, 10, 15, 20, 40, 50, 100, 150, 200} | $\alpha_i = 2$ for $i = 1, \ldots, 10;$ $\alpha_i = 0.2$ for $i = 11, \ldots, 50.$ |

effect on the output. The six-dimensional test function $f_6$ [3, 16] is given by:

$$g_1(x_1) = -\sin(\pi x_1) - 0.3\sin(3.33\pi x_1); \tag{43}$$

$$g_2(x_2) = -0.76\sin(\pi(x_2 - 0.2)) - 0.315; \tag{44}$$

$$\begin{aligned} g_3(x_3) &= -0.12\sin(1.05\pi(x_3 - 0.2)) \\ &\quad -0.02\sin(95.24\pi x_3) - 0.96; \end{aligned} \tag{45}$$

$$g_4(x_4) = -0.12\sin(1.05\pi(x_4 - 0.2)) - 0.96; \tag{46}$$

$$g_5(x_5) = -0.05\sin(\pi(x_5 - 0.2)) - 1.02; \tag{47}$$

$$g_6(x_6) = -1.08; \tag{48}$$

$$f_6(\mathbf{x}) = \sum_{i=1}^{6} g_i(x_i), \tag{49}$$

where $\mathbf{x} \in [0, 1]^6$ uniformly. Note that this model is purely additive, which causes QR radial and winding methods to produce identical results: $f_6(\mathbf{x} + \delta_i) - f_6(\mathbf{x}) = g_i(x_i + \delta_i) - g_i(x_i)$, so effects are the same for a radial and winding design (given the underlying QR sequences are the same). As an example of an environmentally relevant dimensional test case with non-unit input ranges, we consider the Penman-Monteith equation for evapotranspiration [49], given in energy flux rate form by:

$$ET = \frac{\Delta_{ET}A_{ET} + \rho_a c_p g_a \text{VPD}}{\Delta_{ET} + \gamma(1 + g_a/g_s)} \quad [\text{Wm}^{-2}], \tag{50}$$

where $\Delta_{ET}$ is the rate of change of saturation specific humidity with air temperature, $A_{ET}$ is the difference between net irradiance and ground heat flux (i.e. the available energy), $\rho_a$ is the dry air density, $c_p$ is the specific heat capacity of air, VPD denotes the vapor pressure deficit, $g_a$ represents air conductivity, $\gamma$ is the psychromatic constant and $g_s$ represents stomatal conductivity. The units and ranges of the input parameters are listed in Table 5.

**Table 5. Units, input ranges and Sobol total indices $S_{T_i}$ of the input parameters of the Penman-Monteith equation for evapotranspiration (Eq (50)).**

| Param. | Units | $min_i$ | $max_i$ | $S_{T_i}$ | Source |
|---|---|---|---|---|---|
| $\Delta_{ET}$ | kPa C$^{\circ-1}$ | 0.05 | 0.4 | 0.0225 | [50] |
| $A_{ET}$ | W m$^{-2}$ | 0 | 400 | 0.0467 | [51] |
| $\rho_a$ | kg m$^{-3}$ | 1.1 | 1.3 | 0.0081 | |
| $c_p$ | J kg$^{-1}$ C$^{\circ-1}$ | 1000 | 1050 | 0.0007 | |
| VPD | kPa | 0.3 | 3 | 0.6420 | |
| $g_a$ | m s$^{-1}$ | 0.0133 | 0.25 | 0.1108 | [51] |
| $\gamma$ | kPa C$^{\circ-1}$ | 0.065 | 0.07 | 0.0013 | |
| $g_s$ | m s$^{-1}$ | 0.005 | 0.02 | 0.2929 | [51] |

## 5.3 Comparing trajectory generation methods for estimating Sobol total sensitivity indices

Here we compare the ability of different trajectory generation methods to estimate the Sobol total sensitivity index $S_{T_i}$ (see S3 Appendix in S1 File for details) for the *K*-, *G*\*-, and Penman-Monteith functions presented in Section 5.2. While this paper focuses on EE, not Sobol/variance-based SA, this test is valuable because i) analytical values for the Sobol total indices are readily available (or can easily be approximated); and ii) we may expect results to carry over to ranking parameters in the EE framework (S3 Appendix in S1 File). Related work has recently been published by Hoyt and Owen [52], who compare radial and winding schemes in the context of computing the mean dimension (which can be expressed as a sum of Sobol indices).

**Numerical estimation of Sobol indices.**   The $S_{T_i}$ of an output $Y$ are estimated from a set of sample points using the Jansen estimator [53, 54]:

$$\widehat{S}_{T_i}(Y) = \frac{\frac{1}{2r}\sum_{j=1}^{r}\left[Y(A_j) - Y(A_{B_j}^{(i)})\right]^2}{\widehat{V}(Y)}, \tag{51}$$

where $\widehat{V}(Y)$ approximates the total variation, and is given by (following [30]):

$$\widehat{V}(Y) = \frac{1}{2r-1} \cdot$$
$$\cdot \left(\sum_{j=1}^{r}\left[Y(A_j) - Y_0\right]^2 + \sum_{j=1}^{r}\left[Y(B_j) - Y_0\right]^2\right); \tag{52}$$

$$Y_0 = \frac{1}{2r}\left(\sum_{j=1}^{r}Y(A_j) + \sum_{j=1}^{r}Y(B_j)\right). \tag{53}$$

Here $Y(A_j)$ is the value of $Y$ at the $j$-th base point, $Y(B_j)$ is the value of $Y$ at the $j$-th row of $B$, and $Y(A_{B_j}^{(i)})$ is the value of $Y$ at the perturbed value in the $x_i$-direction. The perturbed points are not taken into account for the estimated total variance $\widehat{V}(Y)$, because that would lead to a biased estimate. S3 Appendix in S1 File lists an alternative common estimation procedure for $\widehat{V}(Y)$.

**Test setup.**   Performance of the sampling techniques is measured by the mean absolute error (MAE) of the absolute difference between the estimated (Eq (51)) and analytical Sobol

total sensitivity indices over 50 replications of the full experiment with $r$ trajectories, given by

$$\text{MAE} = \frac{1}{50k} \sum_{j=1}^{50} \sum_{i=1}^{k} \left| \widehat{S}_{T_i} - S_{T_i} \right|. \tag{54}$$

By (full) experiment we mean the set of simulations and corresponding outputs required to calculate all $k$ Sobol total sensitivity indices once. The analytical Sobol total indices $S_{T_i}$ for the $K$- and $G^*$-function are given in the S1 File (Eqs. (S20)–(S23)). For the Penman-Monteith Eq (50), they are approximated using the Sensobol package [55] in R (default settings) on a base sample size of $2^{17}$ (see Table 5). Uniqueness of the replicates is ensured in the following ways. For the $K$-function and Penman-Monteith equation, a different part of the QR-sequence is used in each replicate, i.e. the first replicate uses elements 1 to $r$, the second replicate uses elements $r + 1$ to $2r$, etc. For the $G^*$-function, we use the same part of the QR sequence each time, but randomly sample the values of $\eta_i$ in Eq (42), $i = 1, \ldots, k$, since the total sensitivity index is independent of $\eta_i$. Differences with the experiment in [30] are listed in S3 Appendix in S1 File. For the low-dimensional $K$- and $G^*$-function ($k = 10$), the MAE is calculated for $r = 186, 372, 745$ and $1489$ trajectories, as in [30]. In the higher-dimensional tests (i.e. $K$-function with $k = 75$, $G^*$-function with $k = 50$ inputs), the number of trajectories is lowered to $r \in [3, 100]$ to keep the experiment computationally feasible. For the Penman-Monteith equation, we use $r \in [2, 100]$, allowing inclusion of EOT. It was not computationally feasible to apply EOT in the $K$- and $G^*$-function tests, but its behavior is expected to resemble the 'pinned' Sobol radial method as these methods are very similar (see start of Sec. 5). To assess the variability in estimating $S_{T_i}$ caused by the randomness in the $G^*$-function (as the $\eta_i$'s are randomly sampled), the MAE is calculated five times for $k = 10$ (resulting in a total of $5 \cdot 50 = 250$ replicates of the full experiment) and 9 times for $k = 50$ (450 replicates).

**Results.**    For the **$K$-function** ($k = 10$; Fig 8(a)) differences between our 'standard' Sobol results and Saltelli et al.'s are caused by using improved direction vectors [33] in the Sobol QR sequence (the equivalent of a pseudo random seed). 'Pinned' Sobol radial is not visible in the plot, as the MAEs of this approach were much larger than the plot range ($\approx 0.08$). Our results do not show that a radial design unconditionally outperforms a (Sobol) winding design, which is reported in [30]; instead it depends on the chosen QR sequence. For this test function, sampling techniques with a small step size (QR radial/winding) are better than those with a large step size ('pinned' Sobol, EOT), with MAE values being 7–20 times smaller for small step size methods. Although there is significant variation present in the **$G^*$-function** ($k = 10$, Fig 8(b)), indicating that 50 replicates might be low, it is clear that the 'pinned' method employing a large step size performs worse. Moreover, for this more non-linear function radial designs have lower errors ($25 - 40\%$ lower MAE) than their winding counterparts, which is consistent with [30]. For both the $K$- and $G^*$-function ($k = 10$), the $R_d$ sequence performs similar to or worse than Sobol sequences. In the higher-dimensional tests (i.e. **$K$-function with $k = 75$, $G^*$-function with $k = 50$ inputs**; Fig 9), we discard the $R_d$-based sampling strategies as these performed equal to or worse than their respective equivalent using a Sobol sequence in the lower-dimensional tests. Surprisingly, 'pinned' Sobol radial seems to perform better (up to 70% lower MAE) than 'standard' Sobol winding for the $G^*$-function with 50 inputs (as opposed to the $k = 10$ case), although the standard deviation bounds are large and overlap for low $r$. Nevertheless, 'standard' Sobol radial still results in the lowest MAE (up to 80% lower than 'pinned'). In the 75-dimensional $K$-function differences between 'standard' Sobol radial and 'standard' Sobol winding are negligible, but 'pinned' Sobol radial again shows a significantly higher MAE (up to 10 times). Results for the $K$-function with $k = 50$ and $k = 100$ are not shown, but are similar to the $k = 75$ case. Results for the **Penman-Monteith function** (Fig 10)

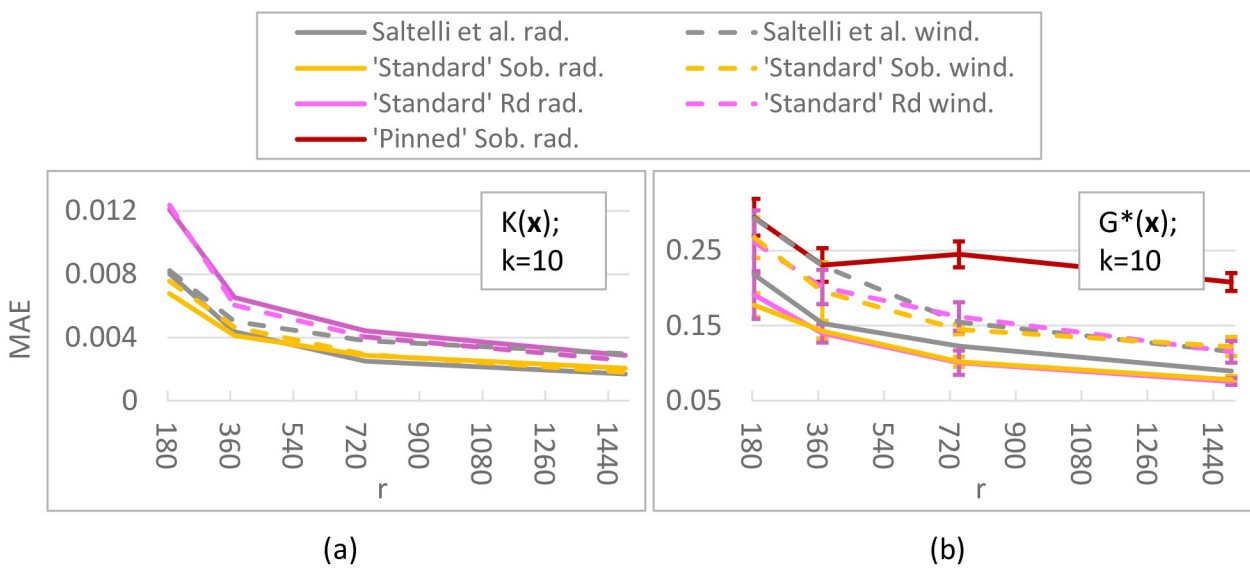

**Fig 8. MAE for Sobol total sensitivity index of $K$-function (a) and $G^*$-function (b); $k = 10$ parameters.** (a): MAE (Eq (54)) of 50 replicates of the complete experiment with $r = 186, 372, 745, 1489$ trajectories applied to the $K$-function (Eq (41)). Green lines show results obtained by Saltelli et al. [30]. In (a), 'pinned' Sobol radial produced larger errors than the plot range shown here. (b): Mean MAE (Eq (54)) ± 1 std bounds of 250 (5 · 50) replicates of the complete experiment applied to the $G^*$-function (Eq (42) and Table 4 with $k = 10$).

show a negligible difference between a radial and winding design and between a Sobol or $R_d$ QR-sequence, but significantly larger errors for large step-size methods ('Pinned' Sobol radial and EOT; up to 20 times higher MAE). Finally, our results suggest **spread** and **discrepancy** are not useful proxies of trajectory generation strategy performance. Comparing the ordering of the different strategies based on spread/discrepancy (Sec. 5.1, Figs 6 and 7) with those based on the MAE (Figs 8–10), it is clear that the orderings are almost opposite in the case of spread, and there is only a partial agreement in the case of discrepancy (for low $k$, 'pinned' methods are correctly estimated to exhibit higher errors).

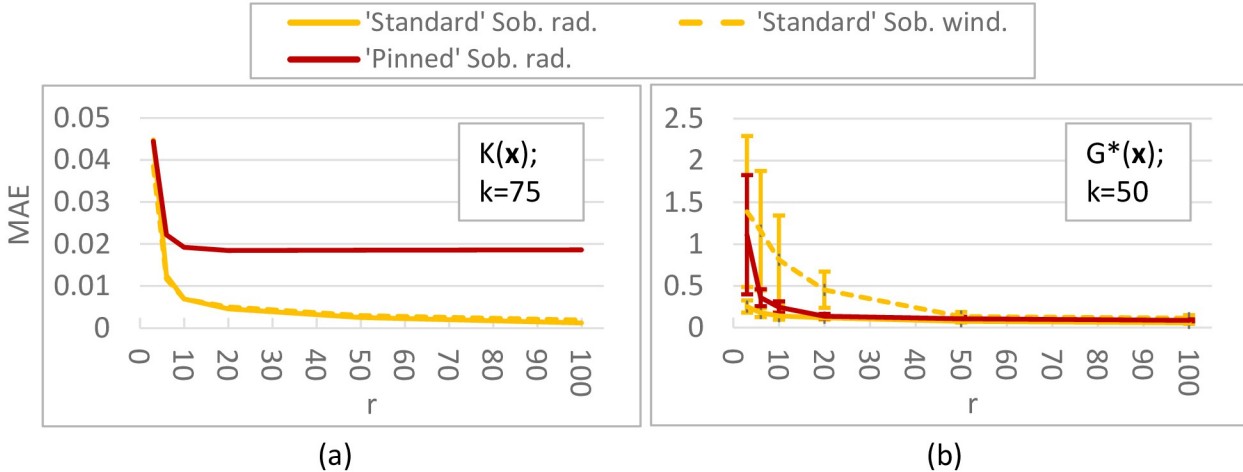

**Fig 9. MAE for Sobol total sensitivity index of K-function ($k = 75$ parameters; (a)) and $G^*$-function ($k = 50$; (b)).** (a): MAE (Eq (54)) of 50 replicates of the complete experiment with $r = 3, 6, 10, 20, 50, 100$ trajectories applied to the $K$-function (Eq (41)). (b): Mean MAE (Eq (54)) ± 1 std bounds of 450 (9 · 50) replicates of the complete experiment applied to the $G^*$-function (Eq (42) and Table 4 with $k = 50$).

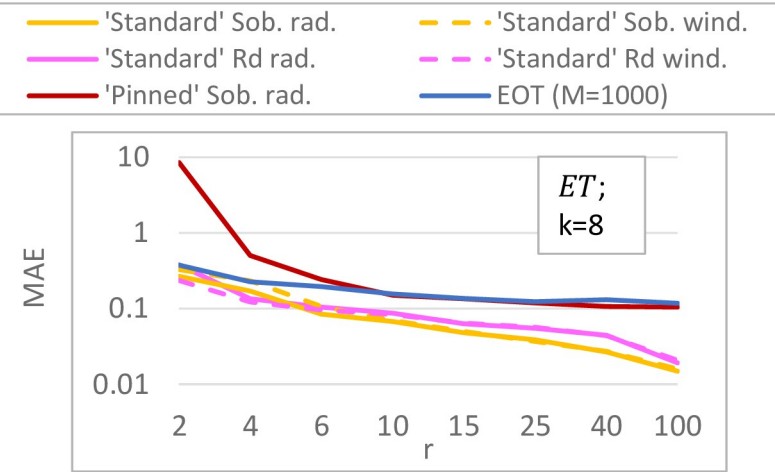

**Fig 10. MAE (Eq (54)) for Sobol total sensitivity indices of the Penman-Monteith function for evapotranspiration (k = 8 parameters; Eq (50)).** MAE is calculated over 50 replicates of the complete experiment with r = 3, 6, 10, 20, 50, 100 trajectories.

## 5.4 Comparing parameter importance rankings from estimated Elementary Effects and rigorous Sobol sensitivity

While it may be expected that the ability to accurately estimate $S_{T_i}$ (of one approach relative to others) generally translates to an equal ability to rank parameters and/or identify (un)important parameters [56], the latter should be tested separately. Furthermore, estimating Sobol sensitivity indices does not give any information about the performance of EE aggregation methods (e.g. $\mu_i^{\star}$ or $\chi_i$). Therefore, we revisit the aforementioned test functions and sampling strategies, but now with a focus on the ability of the EE-based sensitivity measures $S_{\mu^*}$ and $S_{\chi}$ to correctly rank the inputs when compared to Sobol sensitivity rankings.

**Correlation coefficients.** Following [54], the following coefficients are used to ascertain how well the predicted EE-based rankings match with the analytical rankings. The analytical rankings are based on the (approximated) analytical values of the Sobol total sensitivity indices (S3 Appendix in S1 File Eqs. (S20)-(S23); Table 5). Firstly, the Kendall ($\tau$-a) correlation coefficient [57], given by

$$\rho_{\text{kendall}}(\mathbf{x}, \mathbf{y}) = \frac{(\# \text{ concordant pairs}) - (\# \text{ disconcordant pairs})}{\binom{k}{2}}$$
$$= \frac{2}{k(k-1)} \sum_{i<j} \text{sign}(x_i - x_j)\text{sign}(y_i - y_j), \tag{55}$$

where **x** and **y** are sets of observations (in our case analytical and estimated ranks), is a measure of correlation between estimated and actual ranking. It gives equal weight to all ranks. [54] uses the $\tau$-b coefficient, which accounts for ties. Ties do not occur in these rankings, however, so here the simpler $\tau$-a variant is used. For the second coefficient, the ranks are transformed to Savage scores. The score of a parameter with rank $j$ becomes [54]

$$s_j = \sum_{i=j}^{k} \frac{1}{i}, \tag{56}$$

e.g. if $k = 3$ and the parameters are ranked from most to least important as $x_1, x_2, x_3$, the respective Savage scores are $\frac{11}{6}, \frac{5}{6}$ and $\frac{1}{3}$. Subsequently the Pearson correlation coefficient of these transformed quantities (here again denoted by **x** and **y**) is calculated, which is given by [54]

$$\rho_{\text{pearson}}(\mathbf{x}, \mathbf{y}) =$$

$$\frac{n \sum_{i=1}^{n} x_i y_i - \sum_{i=1}^{n} x_i \sum_{i=1}^{n} y_i}{\sqrt{n \sum_{i=1}^{n} x_i^2 - \left(\sum_{i=1}^{n} x_i\right)^2} \sqrt{n \sum_{i=1}^{n} y_i^2 - \left(\sum_{i=1}^{n} y_i\right)^2}}. \tag{57}$$

$\rho_{\text{pearson}}$ assigns more weight to correctly identifying the most important parameters. Both $\rho_{\text{kendall}}$ and $\rho_{\text{pearson}}$ take values in $[-1, 1]$, with 0 indicating no correlation at all, and 1 meaning the estimated and actual rankings are identical.

**Test setup.** Uniqueness of the 50 replicates is obtained as in Section 5.3. The number of trajectories is restricted to practical values, i.e. $r = 2, 4, 6, 10, 15, 25, 40, 100$. EOT is not shown for models with $k = 50$ out of computational considerations.

**Results.** As expected, almost all sampling strategies are capable of accurately ranking the input parameters of the **K-function** (both for $k = 10$ and $k = 50$) with a low number of trajectories (Figs 11(a), 11(b) and 12(a), 12(b)). The only exceptions are EOT and 'pinned' Sobol radial based on the mean of absolute effects $\mu_i^\star$ for $k = 10$, which nevertheless still reach Pearson correlations over 0.9. The case of the **G\*-function** is more interesting (Figs 11(c), 11(d) and 12(c), 12(d)). Approaches based on the median $\chi_i$ generally perform equal or better than their counterparts based on $\mu_i^\star$, especially in ranking important parameters ($k = 10$: $\rho_{\text{pearson}}$ between 2% lower and 57% higher). For $k = 10$ (Fig 11(c) and 11(d)) EOT consistently yields among the lowest correlations, both based on $\mu_i^\star$ and $\chi_i$. Interestingly, 'standard' $R_d$ radial gives among the highest correlations for both $k = 10$ and $k = 50$, while it was inferior in estimating total sensitivity indices. For the **$f_6$-function** the methods employing small step sizes ('standard' Sobol/$R_d$ radial or winding) clearly outperform large step size methods (EOT and 'pinned' Sobol radial) (Fig 13). There are no clear differences between Sobol and $R_d$ QR sequences. Median-based small step-size approaches result in higher Pearson correlations (up to 34%), but roughly equal Kendall correlations. Results for the **Penman-Monteith equation** paint a different picture (Fig 14). Approaches based on the mean $\mu_i^\star$ clearly outperform their counterparts based on the median $\chi_i$. Differences between sampling strategies are small ($S_{\mu^*}$: less than 0.1 difference in $\rho_{\text{kendall}}$ for $r > 10$), although EOT generally results in the lowest correlations and 'Pinned' Sobol radial performs best. Fig 14 also highlights the need for scaling the effects; without scaling, no strategy is capable of accurately ranking the input parameters (all Kendall correlations less than 0.5).

To summarize, small step size methods ('standard') generally perform better than or equal to large step size methods (EOT/'pinned'). There is no one sensitivity measure that always results in the highest correlations; our results indicate $S_{\mu^*}$ (based on the mean of absolute effects) might in some cases be preferable, but in other cases $S_\chi$ (based on the median of absolute effects) yields higher correlations between analytical and calculated rankings. Notably, 'standard' $R_d$ radial performs well across the range of $r$ tested for the $G^*$-function, making it an interesting trajectory generation method for further research. Finally, the results in this section are further proof that **spread** and **discrepancy** are poor proxies of trajectory generation method performance (see Sec. 5.1 and Figs 6 and 7).

## 6 Conclusion

In this work, we looked at the Elementary Effects (EE) sensitivity analysis method in the context of unscaled dimensional models with potentially arbitrary input types (real, integer,

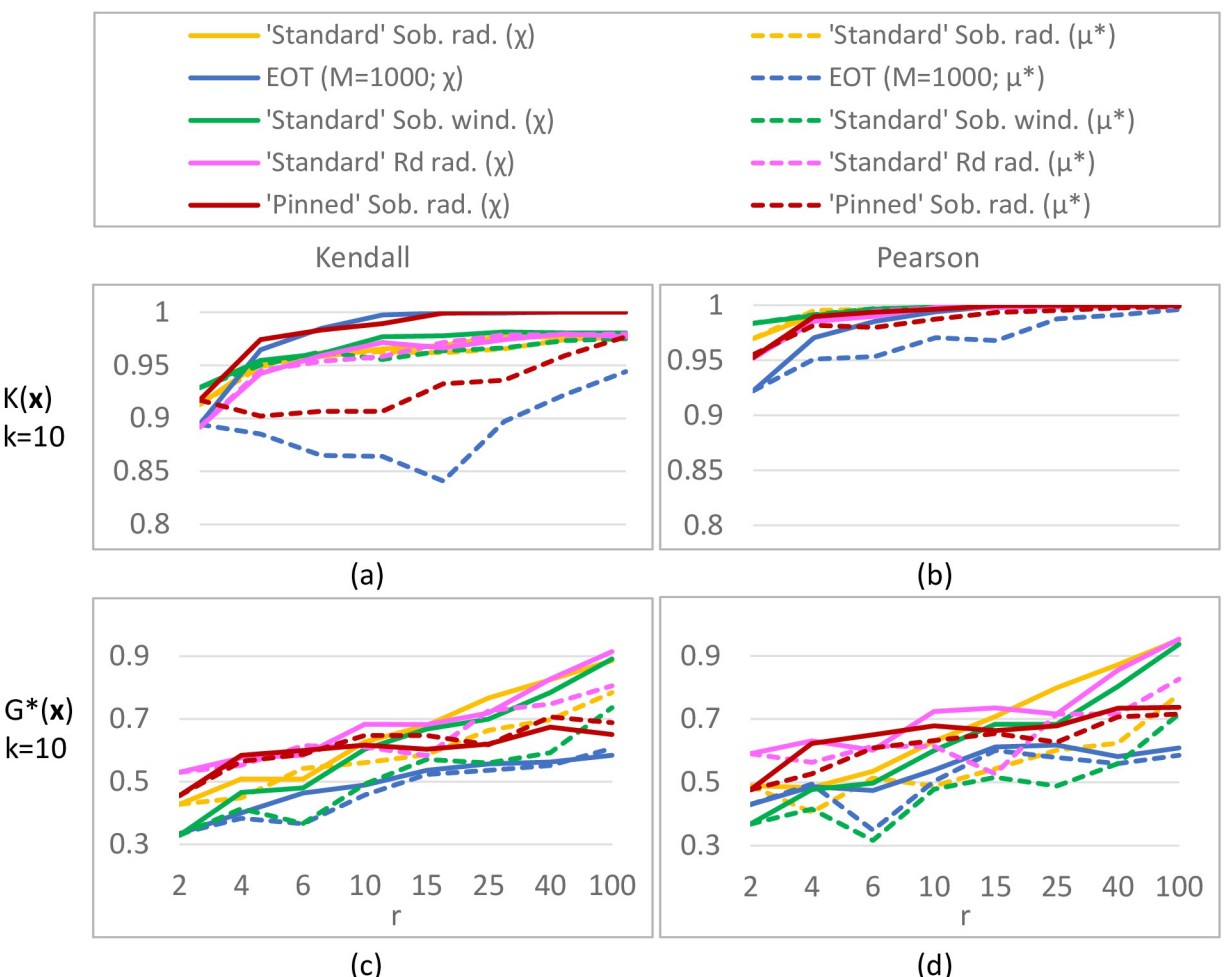

**Fig 11. Correlations** $\rho_{\text{kendall}}$ **(Eq (55)) and** $\rho_{\text{pearson}}$ **(Eq (57)) between estimated and actual parameter rankings for the K-function (Eq (41);** **(a)-(b)) and** $G^*$**-function (Eq (42) and Table 4; (c)-(d)) with k = 10 parameters.** The Pearson correlation assigns more weight to important parameters. The means of the correlation coefficients are shown, based on 50 replicates of the full experiment.

Boolean). We showed that **where model parameters are dimensional or take values on non-unit intervals it is necessary to scale the effects in the input direction by a function of the input parameter range, e.g.** $\max_i - \min_i$**, to avoid erroneous ranking results.** Existing descriptions, software implementations and numerous (including very recent) applications of elementary effects methods do not take scaling or parameter units into account, which may yield results that are wrong. However, scaling by (a function of) the input parameter range has the significant drawback of making the effects directly dependent on the input range, making it of paramount importance to choose parameter bounds with care. Scaling in the output direction is not required to ensure consistent rankings, and is best avoided since these scalings are necessarily dependent on sampled simulation points or experimental data. **We propose two new dimensionless normalized measures based on existing literature (similar to the differential importance measure in [45]):** $S_\chi$ **(Eq (36), based on the median of absolute effects) and** $S_{\mu^*}$ **(Eq (35), based on the mean of absolute effects).** Because the measures are normalized, they allow for a standardized way of identifying (un)important parameters (as described in [40]).

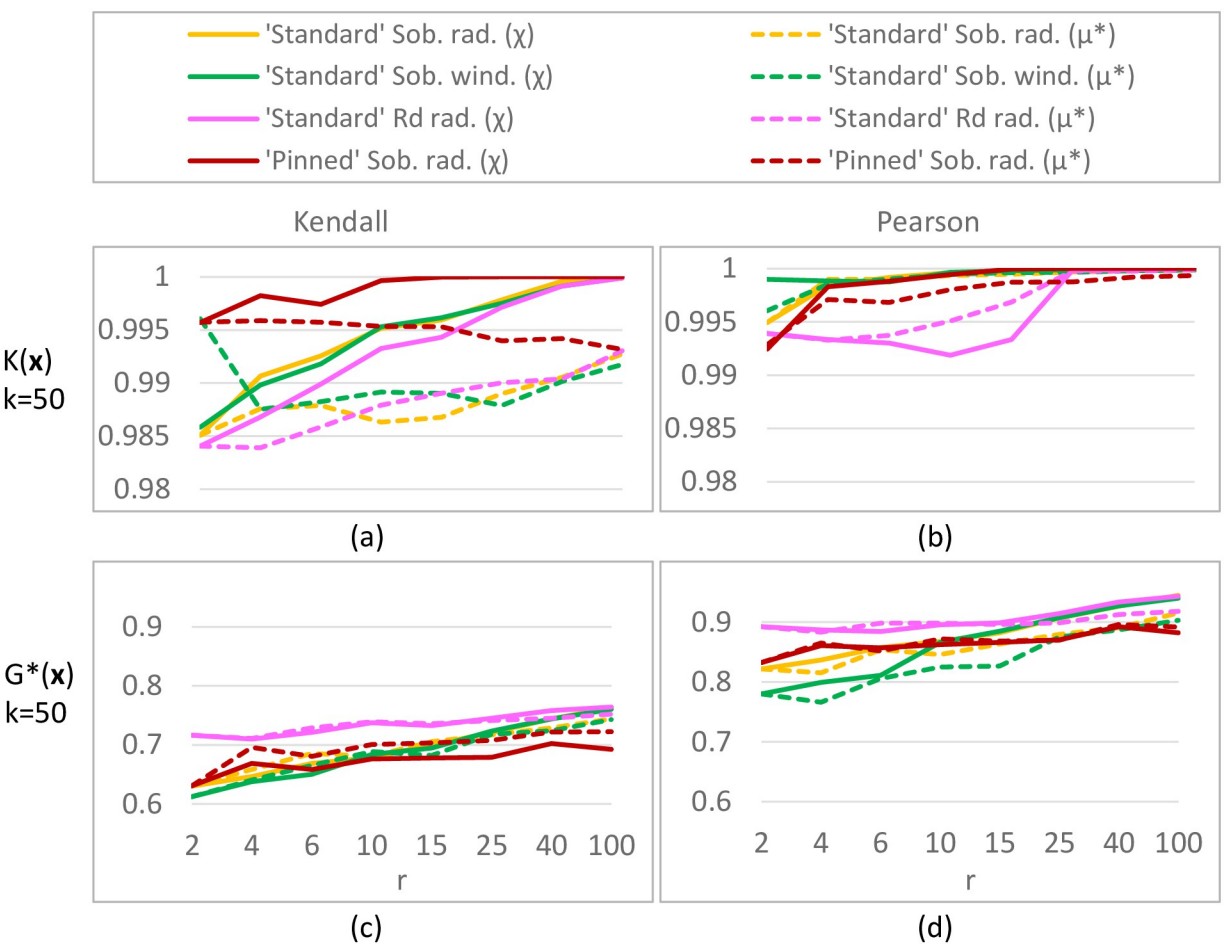

**Fig 12. Correlations $\rho_{\text{kendall}}$ (Eq (55)) and $\rho_{\text{pearson}}$ (Eq (57)) between estimated and actual parameter rankings for the $K$-function (Eq (41); (a)-(b)) and $G^*$-function (Eq (42) and Table 4; (c)-(d)) with $k = 50$ parameters.** The Pearson correlation assigns more weight to important parameters. The means of the correlation coefficients are shown, based on 50 replicates of the full experiment.

We evaluated the ability of 9 trajectory generation methods to calculate Sobol total sensitivity indices for 4 different test functions (extending the experiment in [30] (Figs 8–10)), and subsequently assessed the ability of the new EE-based sensitivity indices (given a trajectory generation method) to rank parameters for the 4 different test functions (Figs 11–14). This revealed:

- Methods employing the mean-based measure $S_{\mu^*}$ (Eq (35)) can perform approximately equal to (Figs 11 and 12), better than (Fig 14), or worse than (Fig 13(c)) those using the median-based measure $S_\chi$ (Eq (36)). In contrast, [28] finds that median-based measures result in more stable ranking results.

- Small step size methods (i.e. those using step sizes dictated by QR sequences) generally perform equal to or better than large step size methods (e.g. EOT, 'pinned' versions) (Figs 8–10 and 13).

- There is no consistent and clear difference between methods employing a Sobol QR sequence versus those using the new $R_d$ sequence. Nevertheless, the performance of the $R_d$ sequence in the $G^*$- and $f_6$-functions (Figs 12(c), 12(d) and 13) in combination with its simple description merits further research into its potential applications.

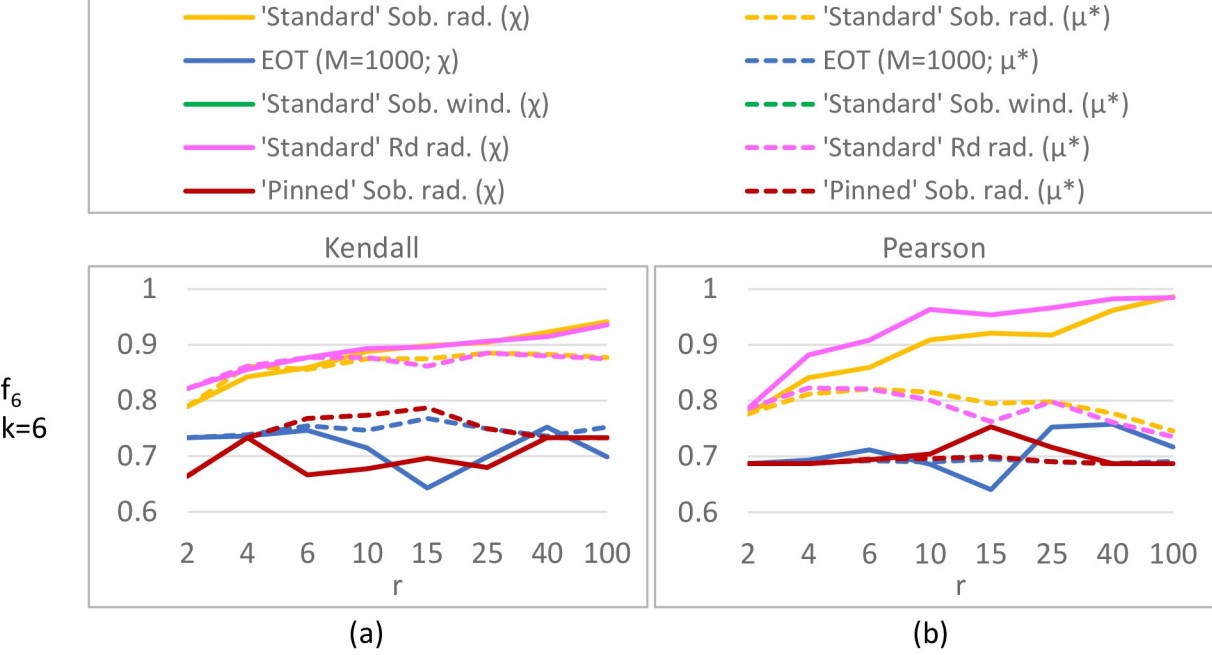

**Fig 13. Correlations** $\rho_{\text{kendall}}$ **(Eq (55)) and** $\rho_{\text{pearson}}$ **(Eq (57)) between estimated and actual parameter rankings for the** $f_6$**-function (Eq (49)).** The Pearson correlation assigns more weight to important parameters. The means of the correlation coefficients are shown, from 50 replicates of the full experiment. Note that 'standard' Sobol radial (yellow) and winding (green) produce identical results (given a sensitivity measure), as the test function is purely additive; only the yellow line is visible.

- While [30] concludes a radial design is the preferred choice over a winding design, our results show no consistent and clear distinction between the two, except in the $G^*$-function (Fig 9(b)) where radial designs are slightly better.

- The Penman-Monteith evapotranspiration example clearly shows the importance of scaling EE-based sensitivity indices to obtain correct rankings (Fig 14).

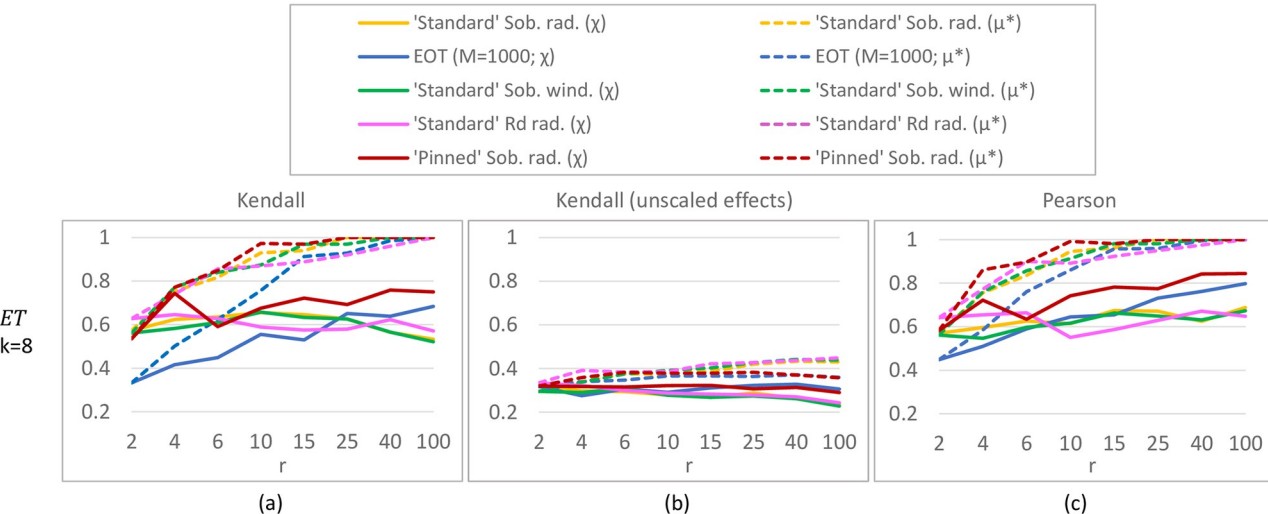

**Fig 14. Correlations** $\rho_{\text{kendall}}$ **(Eq (55)) and** $\rho_{\text{pearson}}$ **(Eq (57)) between estimated and actual parameter rankings for the Penman-Monteith function for evapotranspiration (Eq (50)).** The Pearson correlation assigns more weight to important parameters. The means of the correlation coefficients are shown, from 50 replicates of the full experiment. The middle figure shows $\rho_{\text{kendall}}$ using **unscaled** effects, leading to incorrect rankings.

**Our recommendation is therefore to always compute both sensitivity measures, and to further investigate the output data if the resulting rankings differ significantly. A small step size method is preferred, but it does not seem to be important whether one uses a Sobol or $R_d$ QR sequence, or whether one uses a winding or radial design.**

Finally, we showed that **trajectory spread and discrepancy of the set of simulation points are poor predictors for the ability of a trajectory generation method to correctly rank parameters, identify (un)important inputs or calculate sensitivity indices.** This raises the question of what are good proxies of performance, and whether basing sampling techniques on spread maximisation (e.g. EOT) or discrepancy minimization should be avoided. Recent work by Lo Piano et al. [58] on the trade-off between explorativity (the fraction of non-repeated coordinates in the design) and economy (the number of elementary effects obtained from a given number of simulations) could be an alternative to considering spread and discrepancy, although the designs in our work have both equal explorativity and economy.

In the future, it would be interesting to investigate more QR-sequences than Sobol and $R_d$, and to investigate further the performance of small versus large step size methods. Nevertheless, **this work provides modellers with an up-to-date formulation of EE for general models, thereby aiding model development in the biological and environmental sciences.**

## Supporting information

**S1 File. Appendices.**
(PDF)

**S2 File. Code.** Code (in the XL language) used to obtain the numerical results.
(ZIP)

**S3 File. Additional data and figures.** All data—Including additional figures—In the form of Microsoft Excel spreadsheets.
(XLSX)

**S4 File.**
(PDF)

## Acknowledgments

This work was supported by the University of Nottingham Future Food Beacon of Excellence. We thank Paola Annoni, Jessica Cariboni and Prof. Gürkan Sin for providing additional information regarding their works.

## Author Contributions

**Conceptualization:** Rik J. L. Rutjens, Leah R. Band, Matthew D. Jones, Markus R. Owen.

**Investigation:** Rik J. L. Rutjens.

**Methodology:** Rik J. L. Rutjens, Leah R. Band, Matthew D. Jones, Markus R. Owen.

**Software:** Rik J. L. Rutjens.

**Supervision:** Leah R. Band, Matthew D. Jones, Markus R. Owen.

**Visualization:** Rik J. L. Rutjens.

**Writing – original draft:** Rik J. L. Rutjens.

**Writing – review & editing:** Rik J. L. Rutjens, Leah R. Band, Matthew D. Jones, Markus R. Owen.

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
