## [Decision Letter · Decision Letter 0]

11 Aug 2023

PONE-D-23-20267Elementary Effects for models with dimensional inputs of arbitrary type and range: Scaling and trajectory generationPLOS ONE

Dear Dr. Rutjens,

Thank you for submitting your manuscript to PLOS ONE. After careful consideration, we feel that it has merit but does not fully meet PLOS ONE’s publication criteria as it currently stands. Therefore, we invite you to submit a revised version of the manuscript that addresses the points raised during the review process.

We look forward to receiving your revised manuscript.

Kind regards,

Abel C.H. Chen

Academic Editor

PLOS ONE

Reviewers' comments:

Reviewer's Responses to Questions

**Comments to the Author**

1. Is the manuscript technically sound, and do the data support the conclusions?

Reviewer #1: Yes

2. Has the statistical analysis been performed appropriately and rigorously? 

Reviewer #1: Yes

3. Have the authors made all data underlying the findings in their manuscript fully available?

Reviewer #1: No

4. Is the manuscript presented in an intelligible fashion and written in standard English?

Reviewer #1: Yes

5. Review Comments to the Author

Reviewer #1: 1. The novelty of the paper is adequate enough for possible publication at PLOS ONE.

2. Literature review should be strengthen using new published articles.

3. The contribution and conclusion of the paper can better explore descriptively and quantitatively.

4. Some remarks on main results would be necessary and helpful.

6. PLOS authors have the option to publish the peer review history of their article (what does this mean?). If published, this will include your full peer review and any attached files.

Reviewer #1: No

---

## [Author Response · Author response to Decision Letter 0]

18 Sep 2023

Dear Editor,

Many thanks for considering our manuscript ‘Elementary Effects for models with dimensional inputs of arbitrary type and range: scaling and trajectory generation’. We thank the reviewer for their consideration of this manuscript. We were delighted to see that the reviewer thought that “the novelty of the paper is adequate enough for possible publication”. 

We have carefully considered the reviewer’s suggestions and addressed them as described in detail below. These suggestions have led us to revise the text in several places, including updating the literature review and further clarifying the most important contributions of our work. Changes to the text are highlighted in red. We hope that you find the revised manuscript suitable for publication.

Yours Sincerely,

Rik Rutjens

 

Reviewer #1: 

1. The novelty of the paper is adequate enough for possible publication at PLOS ONE.

We appreciate the reviewer’s appraisal of our work.

2. Literature review should be strengthen using new published articles.

The previously submitted version contains 60 relevant references. We feel that these references adequately describe the state of the field and the most important developments. We do however appreciate that there are a few recent papers that have not been included, in particular on alternative approaches to trajectory generation, that could be included in the paper. The following line has therefore been added to Subsection 2.4 of the manuscript: “For a number of recent approaches (including cluster sampling), the reader is referred to \\cite{EE_Feng2022, EE_Shi2023, EE_Shi_2021, EE_Wu2020, EE_Feng2020} and the references therein.” 

Should the reviewer still feel the literature review lacks key references, more specific direction from them would be helpful.

3. The contribution and conclusion of the paper can better explore descriptively and quantitatively.

To improve readability, we have removed or shortened non-essential lines throughout the manuscript, in particular in the results and conclusion sections (highlighted in red in the revised manuscript). 

In addition, we have added quantitative statements (again highlighted in red) to the results paragraphs of Section 5.3 and 5.4, quantifying the difference in MAE between different trajectory generation methods and the differences in correlation coefficients between different combinations of trajectory generation methods and sensitivity indices. We believe the contributions and conclusions of the paper are now sufficiently supported by quantitative statements.

4. Some remarks on main results would be necessary and helpful.

We have attempted to improve highlighting of, and signposting to, the most important results (in line with the abstract), by making key statements in the conclusion boldface. To keep the manuscript a manageable length, we are wary of adding more text.

---

## [Decision Letter · Decision Letter 1]

11 Oct 2023

Elementary Effects for models with dimensional inputs of arbitrary type and range: Scaling and trajectory generation

PONE-D-23-20267R1

Dear Dr. Rutjens,

We’re pleased to inform you that your manuscript has been judged scientifically suitable for publication and will be formally accepted for publication once it meets all outstanding technical requirements.

Kind regards,

Abel C.H. Chen

Academic Editor

PLOS ONE

Additional Editor Comments (optional):

Reviewers' comments:

Reviewer's Responses to Questions

**Comments to the Author**

1. If the authors have adequately addressed your comments raised in a previous round of review and you feel that this manuscript is now acceptable for publication, you may indicate that here to bypass the “Comments to the Author” section, enter your conflict of interest statement in the “Confidential to Editor” section, and submit your "Accept" recommendation.

Reviewer #1: All comments have been addressed

2. Is the manuscript technically sound, and do the data support the conclusions?

Reviewer #1: Yes

3. Has the statistical analysis been performed appropriately and rigorously? 

Reviewer #1: Yes

4. Have the authors made all data underlying the findings in their manuscript fully available?

Reviewer #1: Yes

5. Is the manuscript presented in an intelligible fashion and written in standard English?

Reviewer #1: Yes

6. Review Comments to the Author

Reviewer #1: (No Response)

7. PLOS authors have the option to publish the peer review history of their article (what does this mean?). If published, this will include your full peer review and any attached files.

Reviewer #1: No

---

## [Editor Report · Acceptance letter]

13 Oct 2023

PONE-D-23-20267R1 

Elementary Effects for models with dimensional inputs of arbitrary type and range: Scaling and trajectory generation 

Dear Dr. Rutjens:

I'm pleased to inform you that your manuscript has been deemed suitable for publication in PLOS ONE. Congratulations! Your manuscript is now with our production department. 

Kind regards, 

on behalf of

Dr. Abel C.H. Chen 

Academic Editor

PLOS ONE